# Assessing Subseasonal Forecast Skill for Use in Predicting US Coastal Inundation Risk

John R. Albers[1], Matthew Newman[1], Magdalena A. Balmaseda[2], William Sweet[1,3], Yan Wang[1,4], and Tongtong Xu[1,4]

[1] NOAA Physical Sciences Laboratory.
[2] European Centre for Medium-Range Weather Forecasts, Reading, UK
[3] NOAA National Ocean Service.
[4] Cooperative Institute for Research in Environmental Sciences University of Colorado Boulder.

*Correspondence to*: John R. Albers (john.albers@noaa.gov)

**Abstract.** Developing predictions of coastal flooding risk on subseasonal timescales (2-6 weeks in advance) is an emerging priority for the National Oceanic and Atmospheric Administration (NOAA). In this study, we assess the ability of two current operational forecast systems, the European Centre for Medium-Range Weather Forecasts Integrated Forecasting System (IFS) and the Centre National de Recherches Météorologiques climate model (CNRM), to make subseasonal ensemble predictions of the non-tidal residual component of coastal water levels at United States coastal gauge stations for the period 2000-2019. These models were chosen because they assimilate satellite altimetry at forecast initialization and attempt to predict the mean sea level, including a global mean component whose absence in other forecast systems complicates assessment of tide gauge reforecast skill. Both forecast systems have skill that exceeds damped persistence for forecast leads through 2-3 weeks, with IFS skill exceeding damped persistence for leads up to six weeks. Post-processing forecasts to include the inverse barometer effect, derived from mean sea level pressure forecasts, improves skill for relatively short forecast leads (1-3 weeks). Accounting for vertical land motion of each gauge primarily improves skill for longer leads (3-6 weeks), especially for the Alaskan and Gulf Coasts; sea-level trends contribute to reforecast skill for both model and persistence forecasts, primarily for the East and Gulf Coasts. Overall, we find that current forecast systems have sufficiently high levels of deterministic and probabilistic skill to be used in support of operational coastal flood guidance on subseasonal timescales.

## 1 Introduction

Over the past several decades, nearly all United States coastal regions have experienced a steady increase in the frequency, extent, and duration of high tide flooding, particularly the East and Gulf Coast regions (Sweet et al. 2018, Sweet et al. 2022). The increased flooding frequency is associated with risks to trillions of dollars of property and infrastructure, as well as risks to coastal ecosystems (Fleming et al. 2018 and references therein). As a result, there is an emerging need to provide high tide flooding outlooks on subseasonal-to-annual timescales (Dusek et al. 2022, NOAA Coastal Inundation Framework).

High tide flooding (HTF) can be defined in terms of water levels exceeding gauge station water level thresholds, where the thresholds are typically determined based on local conditions, including topography, land-cover types, and risk to infrastructure (Sweet et al. 2018; Kavanaugh et al. 2023). The water levels, in turn, can be considered the sum of a tidal component, a sea-level trend component, and a non-tidal residual (Dusek et al. 2022; see also Widlansky et al. 2017). For the West Coast and parts of the East Coast, the tidal component dominates water level in terms of amplitude relative to the non-

tidal residual, either because the tidal range is large or because the non-tidal residual is small, while in other regions, including the Gulf Coast, the non-tidal residual is comparable to the tidal component (Merrifield et al. 2013, Sweet et al. 2014). Still, predicting non-tidal residuals is important for all coastal regions because, as sea levels continue to rise, even small amplitude non-tidal residual anomalies can push high tide water levels past flood thresholds (e.g., Sweet and Park, 2014).

     Currently, NOAA's National Ocean Service issues a monthly HTF outlook for a wide range of stakeholders (NOAA

Coastal Inundation Framework; Dusek et al., 2022) by using damped persistence of current gauge station monthly non-tidal residual anomalies as part of its predicted total water levels. If current operational models could in fact skillfully predict coastal sea level anomalies on subseasonal timescales, specifically by outperforming damped persistence, then incorporating those predictions in the HTF outlook could improve upon its forecast guidance. The non-tidal residual is driven by a multitude of processes operating on timescales ranging from minutes to decades (Sweet et al. 2018, Woodworth et al. 2019, and references

therein). Many of these processes, including daily timescale wind forcing and storm surge, are largely governed by individual weather systems, which are generally not predictable beyond 10-14 days (Lorenz, 1963, 1969; Weber and Mass, 2017; Bauer et al., 2015; Simmons and Hollingsworth, 2002; Zhang et al., 2019). However, sea level anomalies are also correlated with several modes of large-scale climate variability, including the Madden-Julian Oscillation (MJO), the El Niño-Southern Oscillation (ENSO), and the North Atlantic Oscillation (NAO) (Enfield and Allen, 1980; Menéndez and Woodworth, 2010;

Sweet and Zervas, 2011, Ezer and Atkinson, 2014; Sweet and Park, 2014; Valle-Levinson et al., 2017; Han et al., 2019; Amaya et al., 2022; Boucharel et al., 2023; Arcodia et al., 2024; Renkl et al., 2024), which are associated with potentially predictable signals on subseasonal-to-seasonal (S2S) timescales (e.g, Vitart and Molteni, 2010; Barnston et al., 2017; Albers et al. 2021 and references therein). For example, Amaya et al. (2022) recently demonstrated that coastal Kelvin waves related to ENSO are associated with sea surface height anomalies along the West Coast of the United States that can be skillfully predicted.

Whether these climate modes also introduce predictable sea level anomalies along the other coasts of the United States via wind stress and atmospheric pressure anomalies associated with atmospheric teleconnections remains an open question.

     In this study, we assess the skill of current forecast models for predicting non-tidal anomalies at gauge stations on subseasonal timescales. Of course, there is a clear difference in spatial scale between the model grid size, on the order of tens of kilometers, and the point locations of the gauge stations. Here, we will take a relatively simple approach, using the closest

oceanic model grid point to a particular gauge station to make forecasts for that station, and do not consider the additional complication of how best to downscale the model output to much smaller scales (e.g., Long et al. 2023). This approach hinges in part on the assumption that open ocean or near-shore sea level subseasonal anomalies will be representative of the sea level subseasonal anomalies at gauge stations. This is a reasonably well-justified assumption, as many coastal gauge stations are

fairly well-correlated with the nearby open ocean (Vinogradov and Ponte, 2011). However, the connection between open ocean and coastal sea level anomalies varies by region due to many factors, including shelf depth and extent, and eastern versus western boundary ocean dynamics (Hughes et al., 2019; Han et al., 2019). For example, for eastern boundary regions (e.g., the US West Coast), coastal sea level will be influenced by coastally trapped waves propagating poleward from equatorial regions and local direct forcing; for western boundary regions, on the other hand, models need to successfully simulate more processes, including wave signals from all latitudes (Hughes et al. 2019). Gauge stations located on islands, on the other hand, which typically have narrow shelfs, should have sea level variability more closely matching the nearby open ocean (Vinogradov and Ponte, 2011). Coastal shelf depth also effects the relative importance of wind stress versus atmospheric pressure effects; for example, along coasts with shallow shelves, wind stress driven sea level variability is relatively more important than the inverse barometer effect (IBE) on daily to monthly timescales (Woodworth et al. 2019 and references therein).

The two forecast systems evaluated here, the European Centre for Medium Range Weather Forecasting Integrated Forecasting System reforecasts (IFS) and the Centre National de Recherches Météorologiques climate model (CNRM), were chosen in part because they both assimilate ocean altimetry data into their initial conditions, which previous studies have suggested is important to sea level forecast skill (Widlansky et al. 2023; Long et al. 2025). In addition, since the forecast systems are verified against tide gauge observations, we examine the impact of post-processing reforecasts by including both the IBE, computed from the corresponding mean sea level pressure reforecasts, and an estimate of local vertical land motion determined for each gauge location. Model and verification datasets and skill evaluation methods, including deterministic and probabilistic metrics, are outlined in Section 2. This is followed by an evaluation of open ocean sea surface height reforecast skill and coastal gauge station skill for the United States (Section 3.1-3.2), where gauge stations are broken down into four regions: the East Coast, Gulf Coast, West Coast, and Alaska. The seasonality of skill is briefly discussed at the end of Section 3.3, which is followed by a discussion of the implications of the results in the Conclusions.

## 2 Data and verification metrics

### 2.1 Forecast models and verification data

Two dynamical forecast models are considered, the European Centre for Medium Range Weather Forecasting Integrated Forecasting System reforecasts (IFS model versions CY 46R1- 47R1 for the reforecast period 2000-2019, https://www.ecmwf.int/en/publications/ifs-documentation) and the Centre National de Recherches Météorologiques climate model (CNRM model version CM6.1, for the reforecast period 1993-2017, Voldoire et al. 2019). Both models utilize the Nucleus for European Modeling of the Ocean (NEMO, Madec et al., 2017) ocean model run at 1/4° resolution, and assimilate ocean altimetry data into their initial conditions (ECMWF ocean analysis for the IFS and the Mercator-Ocean ocean and sea-ice analysis for the CNRM). Reforecast data, obtained from the S2S Prediction database (Vitart et al. 2017), was only available at 1° resolution, so all calculations, including identifying the closest model grid point to each gauge station, is based on this resolution.

For the open ocean, reforecasts are verified against sea surface height (SSH) from the Copernicus Marine Service Global Ocean Physics Reanalysis 12v1 (GLORYS, Lellouche et al., 2021), which has been extensively verified in North American waters (Amaya et al. 2022; Amaya et al. 2023; Castillo-Trujillo et al. 2023; Feng et al. 2024). For coastal regions of the continental United States and Alaska, reforecasts are further verified against National Oceanic and Atmospheric Administration (NOAA) water level gauge stations (NOAA Tides and Currents https://tidesandcurrents.noaa.gov/PageHelp.html). Only gauge stations that have at least 10 years of data during the reforecast records of the IFS and CNRM are considered, yielding 47 stations for the East Coast, 32 stations for the Gulf of Mexico, 35 stations for the West Coast, and 23 stations for Alaska (see Table 1 for a list of the gauge stations and the number of days in each verification time series). When comparing the reforecast skill of the IFS and CNRM (Section 3.1), we verify against their common reforecast periods, 2000-2017; when evaluating the reforecast skill of the IFS alone (Sections 3.2-3.3), we use the full IFS reforecast period, 2000-2019.

As discussed in the Introduction, water levels at gauge stations are separated into three components for the HTF outlooks (Dusek et al., 2022): a 37-constituent tidal component (e.g., Sweet and Zervas, 2011), a linear sea level trend component, and a non-tidal residual. Here, for comparison with the reforecasts, we remove the local tidal components, but we do not remove the trend from the water level gauge data, so that the verification time series of non-tidal residuals ($NTR$) at each tide gauge includes the local trend component, in contrast with how it is defined for the HTF framework. Including the trend component in the reforecasts and verifications of the non-tidal residual allows an assessment of contribution of the linear trend to reforecast skill, for example, by comparing the difference in anomaly correlation skill with and without the linear trend included (Figures 2d, 4d, 6d, and 8d), and comparing reliability with and without the linear trend included (Figures 3, 5, 7, and 9).

IFS reforecasts are available from initialization out to forecast lead day 46, while CNRM reforecasts are available out to forecast lead day 47; both reforecast sets were obtained as daily averages. To create reforecast anomalies, the lead-dependent 20- or 25-year reforecast climatologies of the IFS and CNRM periods, respectively, are substracted from the daily average data, which implicitly applies a mean bias correction; further details regarding the creation of the IFS and CNRM climatologies (which are different for each model) are in the Supplement (Section S1). Next, weekly averages are calculated from the daily reforecast data, where Week 1 includes the average of forecast days 1-7, Week 2 includes days 8-14, etc. To calculate the daily average GLORYS verification anomalies, we use a period spanning both the CNRM and IFS reforecast records (2000-2017), where a 365-day (plus one day for leap years) climatology is calculated by averaging over all years, and extracting the first four harmonics (plus the mean) via Fourier transform (e.g., Epstein, 1988), yielding the final climatology, which is then removed from the daily average GLORYS data. Weekly average GLORYS verification anomalies are then created by applying a 7-day running mean to the daily data. To calculate the NOAA gauge stations anomalies, we calculate a daily average climatology for the period over which the gauge station data is available, generally 20 years but slightly shorter for some stations (see Table 1), and remove that climatology from the daily averaged gauge station data. A 7-day running mean is then applied to the gauge station data.

| East Coast | | West Coast | |
|---|---|---|---|
| **Station list** | **Data length (days)** | **Station list** | **Data length (days)** |
| Eastport, ME (8410140) | 2088 | San Diego, San Diego Bay, CA* (9410170) | 2098 |
| Bar Harbor, ME* (8413320) | 1943 | La Jolla, CA* (9410230) | 2098 |
| Portland, ME* (8418150) | 2097 | Los Angeles, CA* (9410660) | 2098 |
| Boston, MA* (8443970) | 2092 | Santa Monica, CA* (9410840) | 2095 |
| Fall River, MA (8447386) | 2097 | Santa Barbara, CA (9411340) | 1534 |
| Woods Hole, MA (8447930) | 2064 | Oil Platform Harvest, CA (9411406) | 1745 |
| Nantucket Island, MA (8449130) | 2083 | Port San Luis, CA* (9412110) | 2096 |
| Newport, RI* (8452660) | 2098 | Monterey, CA* (9413450) | 2098 |
| Providence, RI* (8454000) | 2086 | San Francisco, CA (9414290) | 2093 |
| New London, CT (8461490) | 2088 | Redwood City, CA (9414523) | 2098 |
| New Haven, CT (8465705) | 2092 | Alameda, CA* (9414750) | 2098 |
| Bridgeport, CT* (8467150) | 2087 | Richmond, CA (9414863) | 2014 |
| Montauk, NY (8510560) | 2003 | Point Reyes, CA* (9415020) | 2088 |
| Kings Point, NY* (8516945) | 2097 | Port Chicago, CA (9415144) | 2081 |
| The Battery, NY* (8518750) | 2032 | Arena Cove, CA* (9416841) | 2098 |
| Bergen Point West Reach, NY* (8519483) | 2049 | North Spit, CA* (9418767) | 2090 |
| Sandy Hook, NJ* (8531680) | 2087 | Crescent City, CA (9419750) | 2064 |

| | | | |
|---|---|---|---|
| Atlantic City, NJ* (8534720) | 2050 | Port Orford, OR* (9431647) | 1847 |
| Cape May, NJ* (8536110) | 2066 | Charleston, OR* (9432780) | 2098 |
| Burlington, Delaware River, NJ (8539094) | 1835 | South Beach, OR* (9435380) | 2098 |
| Marcus Hook, PA (8540433) | 1667 | Garibaldi, OR (9437540) | 1515 |
| Philadelphia, PA (8545240) | 2090 | Astoria, OR (9439040) | 2098 |
| Delaware City, DE (8551762) | 1880 | Wauna, OR (9439099) | 1718 |
| Reedy Point, DE (8551910) | 2076 | St Helens, OR (9439201) | 1863 |
| Lewes, DE* (8557380) | 2093 | Vancouver, WA (9440083) | 1815 |
| Ocean City Inlet, MD (8570283) | 1825 | Skamokawa, WA (9440569) | 1703 |
| Cambridge, MD (8571892) | 2092 | Toke Point, WA* (9440910) | 2058 |
| Tolchester Beach, MD (8573364) | 2093 | Westport, WA (9441102) | 1447 |
| Chesapeake City, MD (8573927) | 1716 | La Push, Quillayute River, WA (9442396) | 1641 |
| Baltimore, MD (8574680) | 2098 | Neah Bay, WA (9443090) | 2096 |
| Annapolis, MD (8575512) | 2065 | Port Angeles, WA* (9444090) | 2098 |
| Solomons Island, MD (8577330) | 2027 | Port Townsend, WA* (9444900) | 2098 |
| Washington, DC (8594900) | 1991 | Seattle, WA* (9447130) | 2098 |
| Wachapreague, VA* (8631044) | 1719 | Cherry Point, WA* (9449424) | 2079 |
| Kiptopeke, VA* (8632200) | 2098 | Friday Harbor, WA* (9449880) | 2098 |
| Lewisetta, VA* (8635750) | 2098 | | |
| Windmill Point, VA* (8636580) | 2048 | | |

| Station list | Data length (days) | | |
|---|---|---|---|
| Yorktown USCG Training Center, VA (8637689) | 1658 | | |
| Sewells Point, VA* (8638610) | 2098 | | |
| Money Point, VA (8639348) | 2096 | | |
| Duck, NC* (8651370) | 2087 | | |
| Oregon Inlet Marina, NC (8652587) | 2089 | | |
| Beaufort, Duke Marine Lab, NC* (8656483) | 2098 | | |
| Wilmington, NC (8658120) | 2083 | | |
| Springmaid Pier, SC* (8661070) | 1962 | | |
| Charleston, SC* (8665530) | 2098 | | |
| Fort Pulaski, GA* (8670870) | 2092 | | |

## Gulf                                    Alaska

| Station list | Data length (days) | Station list | Data length (days) |
|---|---|---|---|
| Fernandina Beach, FL* (8720030) | 2076 | Ketchikan, AK (9450460) | 2098 |
| Mayport (Bar Pilots Dock), FL* (8720218) | 2001 | Port Alexander, AK (9451054) | 1289 |
| Trident Pier, Port Canaveral, FL* (8721604) | 2093 | Sitka, AK (9451600) | 2098 |
| Virginia Key, Biscayne Bay, FL* (8723214) | 2096 | Juneau, AK (9452210) | 2093 |
| Vaca Key, Florida Bay, FL (8723970) | 2065 | Skagway, Taiya Inlet, AK (9452400) | 2087 |
| Key West, FL (8724580) | 2086 | Elfin Cove, AK (9452634) | 1510 |
| Naples, Gulf of Mexico, FL* (8725110) | 2044 | Yakutat, Yakutat Bay, AK (9453220) | 2082 |

| | | | |
|---|---|---|---|
| Fort Myers, FL (8725520) | 2055 | Cordova, AK (9454050) | 2091 |
| Port Manatee, FL (8726384) | 2062 | Valdez, AK (9454240) | 2084 |
| St. Petersburg, Tampa Bay, FL (8726520) | 2095 | Seward, AK (9455090) | 2098 |
| Old Port Tampa, FL (8726607) | 1915 | Seldovia, AK (9455500) | 2074 |
| Clearwater Beach, FL (8726724) | 2067 | Nikiski, AK (9455760) | 2094 |
| Cedar Key, FL* (8727520) | 2047 | Anchorage, AK (9455920) | 2087 |
| Apalachicola, FL (8728690) | 2084 | Kodiak Island, AK (9457292) | 2091 |
| Panama City, FL (8729108) | 2061 | Alitak, AK (9457804) | 1398 |
| Panama City Beach, FL (8729210) | 1496 | Sand Point, AK (9459450) | 2072 |
| Pensacola, FL* (8729840) | 2028 | King Cove, AK (9459881) | 1506 |
| Dauphin Island, AL* (8735180) | 1851 | Adak Island, AK (9461380) | 2051 |
| Mobile State Docks, AL (8737048) | 1387 | Unalaska, AK (9462620) | 2094 |
| Bay Waveland Yacht Club, MS* (8747437) | 1476 | Port Moller, AK (9463502) | 1124 |
| Shell Beach, LA (8761305) | 1191 | Village Cove, St Paul Island, AK (9464212) | 1409 |
| Grand Isle, LA (8761724) | 2062 | Nome, Norton Sound, AK (9468756) | 2067 |
| New Canal Station, LA (8761927) | 1484 | Prudhoe Bay, AK (9497645) | 2090 |
| Port Fourchon, Belle Pass, LA (8762075) | 1684 | | |
| Berwick, Atchafalaya River, LA (8764044) | 1673 | | |
| Lake Charles, LA (8767816) | 1555 | | |
| Calcasieu Pass, LA (8768094) | 1709 | | |

| | | |
|---|---|---|
| Morgans Point, Barbours Cut, TX* (8770613) | 2091 | |
| Eagle Point, Galveston Bay, TX* (8771013) | 2095 | |
| Galveston Pier 21, TX* (8771450) | 2098 | |
| Rockport, TX* (8774770) | 2013 | |
| Port Isabel, TX* (8779770) | 2093 | |

**Table 1**. List of NOAA gauge stations organized by four regions: East Coast, Gulf Coast, West Coast, and Alaska. The number of days listed for each gauge station corresponds to the number of reforecast days used when calculating IFS reforecast skill; the IFS is initialized twice weekly, which means that ~2000 days equates to roughly ~20 years. Gauge stations with an * next to them are stations that are included in the official NOAA High Tide Flooding Monthly Outlooks (Kavanaugh et al. 2023).

**2.2 Inverse barometer effect and vertical land motion**

The satellite altimeter-derived sea level products used to initialize the IFS are processed so as to remove static and high frequency (~20 day cutoff) dynamic atmospheric pressure effects (Ponte and Ray, 2002; Centre national d'études spatiales, 2020). As a result, when the ocean model (e.g., NEMO in the case of the IFS) is run for forecasting purposes, it assumes an atmosphere with no mass (both because the assimilated observations are corrected to remove pressure effects, and because the 'atmospheric pressure' subroutine in NEMO is turned off), so that the SSH reforecasts include neither static nor dynamic responses to atmospheric pressure fluctuations (Tai, 1993). However, the static effect of atmospheric pressure on the ocean surface can be approximated via the so-called inverse barometer effect (Ross, 1854; Doodson, 1923; Groves and Hannan, 1968; Tai, 1993; Arbic, 2005; Ponte, 1992, 2006; Oddo et al., 2014; Long et al., 2021; Feng et al., 2024), which assumes a static ocean response to atmospheric pressure forcing (Tai, 1993; Wunsch and Stammer, 1997). The IBE ($\eta_{ibe}$) is written as:

$$\eta_{ibe} = -\frac{p_{msl} - \overline{p_{msl}}}{\rho_{ocean}\, g} \tag{1}$$

(e.g., Piecuch and Ponte, 2015), where $p_{msl}$ is atmospheric mean sea level pressure, $\overline{p_{msl}}$ is the global mean sea level pressure (MSLP, ocean-only), $\rho_{ocean}$ is the ocean density (assumed to be a constant value of 1025 kg m-3), and $g$ is the acceleration due to gravity. Although the assumptions inherent to the inverse barometer approximation are not always strictly valid, which can lead to deviations from a purely static response (e.g., Wunsch, 1991; Le Traon and Gauzelin, 1997), previous studies have suggested that including the IBE is important on subseasonal-to-seasonal timescales (e.g., Woodworth et al, 2019; Long et al., 2021, Feng et al. 2024). In this study, for the IFS we use the predicted $p_{msl}$ to predict the $\eta_{ibe}$ from (1) at different forecast

leads, which is added to the corresponding SSH reforecasts as a post-processing step for each tide gauge. (For simplicity, we do not display IBE-corrected CNRM reforecasts, as the IFS is generally more skillful.)

Over the course of several decades, the sea level at local gauge stations can change from vertical land motion (VLM) due to a wide range of phenomena, including glacial isostatic rebound and groundwater and/or fossil fuel removal (Larsen et al., 2004; Hu and Freymueller, 2019; Sweet et al. 2022; https://www.climate.gov/news-features/features/interactive-map-how-has-local-sea-level-united-states-changed-over-time; Oelsmann et al., 2024). While VLM rate is too small to impact an individual subseasonal forecast, it could be large enough in some regions to impact long-term skill assessment, and is implicitly

included in a persistence forecast. There exist many methods for estimating VLM (e.g., Kopp et al. 2014; Hammond et al. 2021; Oelsmann et al. 2024), however, here we use a relative simple approach that applies VLM rates as a trend correction to each IFS forecast anomaly time series. The VLM rates we use are provided on a 1°x1° grid (https://oceanservice.noaa.gov/hazards/sealevelrise/Sea_Level_Rise_Datasets_2022.zip; Sweet et al. 2022), and the grid point nearest to each gauge station is used as the rate constant for the entire reforecast period.

In the remainder of the paper, we will refer to two types of reforecasts: (1) IFS and CNRM SSH reforecast anomalies, which are the original ocean model output; and (2) IFS reforecast anomalies that are IBE- and VLM-corrected, which will be referred to as non-tidal residual reforecasts:

$$\widehat{NTR} = \text{SSH} + \text{IBE} + \text{VLM} \qquad (2)$$

where in (2), the SSH and IBE are IFS quantities and the VLM is the trend correction described above. While we could have

adjusted the tide gauge NTR using the VLM prior to evaluating skill, we prefer adding both the IFS-IBE and VLM to the IFS-based SSH reforecast, since $\widehat{NTR}$ is the hindcast that should be verified directly against the NTR that is observed at the tide gauge. GLORYS does not include the IBE, so verifying the IFS and CNRM against GLORYS is done using SSH only.

## 2.3 Skill metrics

Reforecast skill is evaluated both deterministically, using anomaly correlation, and probabilistically, using reliability and

sharpness (Atger, 1999; Jolliffe and Stephenson, 2011; Wilks 2011). Reliability is computed for events that exceed the upper tercile of each gauge station's water level, where the tercile threshold is calculated separately for the water level (non-tidal residual) distributions of the reforecasts ($\widehat{NTR}$) and verifications ($NTR$). Using the tercile from the reforecast ensemble sample distribution for computing the reforecast probability of an event (as opposed to using the observed tercile) amounts to an in-sample bias correction of the reforecast probability distribution (Weisheimer and Palmer 2014), which ensures that the

reforecast probabilities and observed frequencies of tercile events are the same for each gauge station; this correction is needed because, in general, the reforecast water level distributions are underdispersive (i.e., more narrow than observed). For all reliability diagrams, the observed distributions and relative forecast frequencies are split into ten bins (0-10%, 10-20%,…, 90-100%).

To condense reliability metrics onto a single map, we use the slope of a regression line fitted to each gauge station's reliability curve, computed via a weighted least-squares regression fitted to the reliability bin values (i.e., the observed relative

frequency-reforecast probability pairs), where the weights are determined by the number of events in each observed frequency-reforecast probability bin. Since we have applied the reforecast probability bias correction, the regression lines always intersect the tercile-tercile point (0.33, 0.33) on the reliability diagrams, so that a regression line with a slope greater than 0.5 contributes positively to the Brier skill score (e.g., Mason 2004), though in general, any reliability regression line that is relatively close to 0.5 is probably 'useful' for making forecast guidance (see e.g., Weisheimer and Palmer, 2014 for a discussion on this point). We also determine forecast sharpness, which measures the ability of a forecast system to issue more definitive guidance (i.e., beyond simply forecasting the climatological probability), as

$$SHP = \frac{1}{n}\sum_{i=1}^{n} p_i(1)\big(1 - p_i((1)\big) * 400 \tag{3}$$

(Daan 1984; Potts 2011), where $p_i(1)$ is the probability of exceeding the tercile threshold for each forecast, $i$, where each '$i$' refers to a forecast initialization and verification pair. This expression yields $SHP =0$ for the sharpest forecasts (when all the forecasts are either 0% or 100%) and $SHP =100$ for the most blurred forecasts (multiplying the expression in (3) by 400 is not strictly necessary, but is done to provide a more intuitive range of $SHP$ values, spanning 0 to 100 instead of 0 to 0.25). Since a forecast system can be perfectly sharp by always forecasting 0%, which would not be terribly useful, we also report the percentage of forecasts that are in the 90-100% probability bin relative to the 0-10% bin.

Dynamical model reforecast skill is also compared to the skill of "damped" persistence of the observed coastal station anomalies. Each damped persistence forecast is calculated using the mean of the previous seven days, multiplied by the lead-dependent autocorrelation value, where the autocorrelation function is calculated independently from the $NTR$ time series for each gauge station. For all reforecasts, including from the CNRM, the IFS, and damped persistence, reforecasts are compared with and without a linear trend removed from the anomaly time series, where the trend is independently computed relative to the length of each time series being compared.

## 3 Results

We begin with a simple evaluation of the geographic distribution of year-round Week 3 SSH prediction skill for the IFS and CNRM, verifying against GLORYS SSH anomalies, which also do not include the IBE (Section 3.1). Thereafter, we conduct a more detailed regional prediction skill assessment of $\widehat{NTR}$ based on the IFS reforecasts, including evaluating the impact of the postprocessed IBE- and VLM-based reforecast corrections, for gauge stations grouped into four broad regions: the East Coast (Maine to South Carolina), the Gulf Coast (Florida to Texas, and one station in Georgia), the West Coast (California to Washington), and Alaska (Section 3.2). Finally, we evaluate the seasonal dependence of $\widehat{NTR}$ prediction skill (Section 3.3).

### 3.1 Multi-model comparison of  coastal ocean and tide gauge SSH reforecast skill

For the oceans surrounding North America, the geographic distribution of Week 3 SSH anomaly correlation skill for the IFS and CNRM are qualitatively similar, with both models exhibiting relatively higher skill in the eastern portion of the Pacific

Ocean, along the southern coast of Alaska, and the Beaufort Sea, while exhibiting relatively lower skill along the East Coast of the US (Fig. 1). In general, the IFS is more skillful than the CNRM in most regions, with the exception of a small region northeast of the Bahamas, the central portion of the Gulf of Mexico, and portions of the Labrador Sea and Baffin Bay.

For both models, SSH reforecast skill evaluated at the gauge stations tends to correspond reasonably well with nearby open ocean SSH reforecast skill evaluated using GLORYS gridded SSH anomalies for Week 3 (cf. colored shading for the oceans surrounding North America and colored dots for the tide gauges used in this study in Fig. 1). This good comparison justifies the use of nearest neighbor open ocean SSH forecasts to predict tide gauge anomalies in this study. For example, for both tide gauges and the near shore open ocean, reforecast skill for both models is relatively high in Southern California and Maine and relatively low between Virginia and New York (cf. both Figs. 1a and 1b).

For the US coastlines nearest to the gauge stations that are of central interest here, the IFS has superior skill at all forecast leads (see Supplement Figs. S1-S6). While the IFS and CNRM have qualitatively similar skill characteristics, only the IFS SSH has skill that is as good or better than persistence for nearly all contiguous US (CONUS) gauge stations (Figs. S1-S6) as well as many Alaskan stations (not shown), with the exception of those situated in regions far up inland rivers, including stations such as Vancouver, WA or Berwick, Atchafalaya River, LA, which are all far removed (and physically

disconnected) from the nearest IFS or CNRM open ocean grid points.

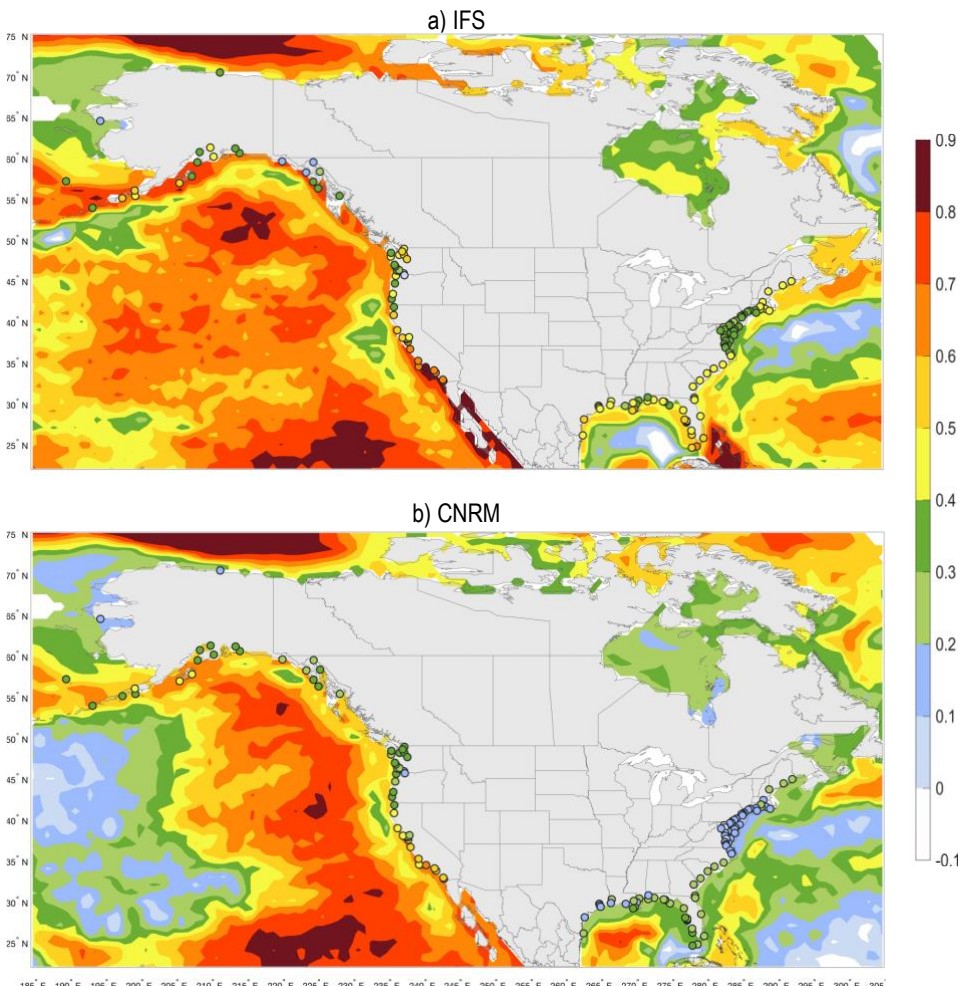

**Figure 1**. Year-round (2000-2017) Week 3 anomaly correlation skill between GLORYS SSH anomalies and SSH reforecasts (color contours), and Week 3 anomaly correlation skill between NOAA gauge station anomalies and SSH reforecasts (colored circle markers), for reforecasts from the (**a**) IFS and (**b**) CNRM. While the sample years used in the anomaly correlation calculation (2000-2017) are the same, the dates are slightly different because of different initialization dates. All dates are used in both datasets, regardless of whether they overlap with the other dataset, because otherwise there are too few samples.

The biggest skill improvements of IFS Week 3 reforecasts relative to persistence are seen in three regions: along the central East Coast, the entire West Coast, and along the southern coast and islands of Alaska (not shown). Reforecast skill is highest in Southern California (e.g., Fig. S3), where skill from coastal Kelvin waves may provide a significant predictable signal (Amaya et al. 2022), particularly during strong El Niño events (see also Arcodia et al., 2024 and references therein). At Week 3, and even at Week 2 (both shown in the Supplement), the CNRM forecasts are uniformly (across all regions) less

skillful than persistence, with one exception: the Week 2 CNRM forecasts for the Pacific Northwest (Oregon and Washington) are more skillful than persistence (Fig. S2).

Scoring reforecasts without accounting for strong trends can make it difficult to differentiate between forecast skill associated with predicting subseasonal climate variations versus the spurious impact of trends on subseasonal forecast skill (Wulff et al. 2022). Similarly, previous studies of seasonal prediction of U.S. coastal sea-level anomalies have suggested that the sea level trend can dominate estimates of reforecast skill, especially when using standard skill metrics that compare to skill of a climatological forecast (Widlansky et al. 2017; Long et al. 2021; Shin and Newman 2021; Long et al. 2025). After a linear trend is removed from the forecast and verification datasets, CONUS reforecast skill is reduced for most regions both for models and for persistence, with the largest effects occurring for leads of 3 weeks and beyond (Figs. S1-S12; see Sections 3.2.1-3.2.4 for a detailed discussion of the effect of the linear trend on IFS reforecast skill). However, removing the linear trend from the model reforecasts and persistence also highlights the models' ability to skillfully predict SSH anomalies related to subseasonal climate variability. For example, linearly detrended CNRM reforecasts are more skillful than linearly detrended persistence for the entire West Coast and most East Coast stations at Weeks 2 and 3 (Figs. S4 and S6, respectively). Thus, with the linear trend removed, it becomes clearer that the CNRM provides useful guidance for many regions for forecast leads out to at least 3 weeks. However, the IFS is still more skillful than the CNRM at nearly all gauge stations and at all lead times (Figs. S1-S12), so for the remainder of the manuscript, we focus on results using the IFS.

**3.2 Regional U.S. coastal skill**

We next assess deterministic and probabilistic $\widehat{NTR}$ reforecast skill for each of the four US subregions, the West, Gulf, and East Coasts, and Alaska. When comparing deterministic skill (anomaly correlation), four panels are shown: $\widehat{NTR}$ skill, followed by a panel showing the difference between $\widehat{NTR}$ skill and persistence skill, and then two panels isolating the contributions of the IBE and the linear trend to $\widehat{NTR}$ skill; the contribution of VLM to $\widehat{NTR}$ skill is discussed in the text when relevant with a figure included in the Supplement.

Probabilistic Week 3 $\widehat{NTR}$ reforecast skill is assessed at all gauge stations via three metrics that characterize reliability and sharpness (see Section 2.3 for details of the calculations): (1) the slope of the reliability regression line, where values greater than 0.5 indicate positive contributions to the Brier skill score; (2) forecast sharpness as measured by the '$SHP$' parameter ranging from 0-100, where smaller values represent sharper forecasts; and (3) the percentage of forecasts that are in the top forecast probability category versus the lowest category (0-10% versus 90-100%), where higher percentages indicate relatively more "certain" affirmative forecasts. In addition, for each sub-region, we show reliability diagrams and histograms of the forecast probability distributions for two representative gauge stations to provide a visual reference for the metric numbers reported on the maps; for these figures, skill of $\widehat{NTR}$ reforecasts with the linear trend removed is also shown.

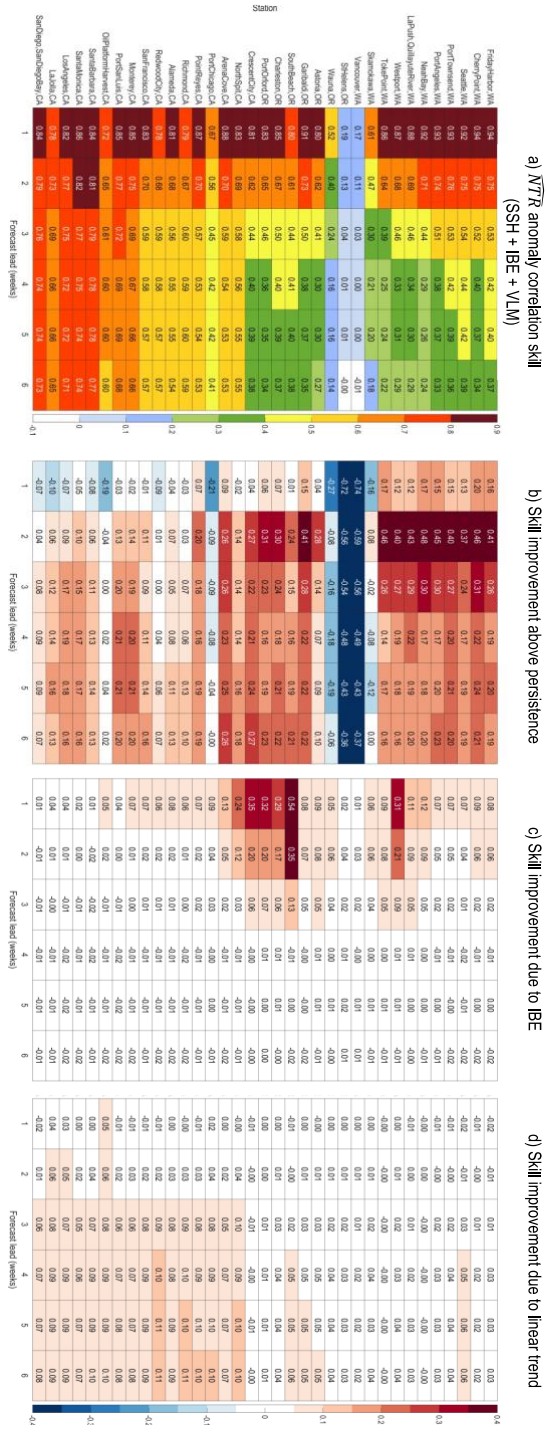

**Figure 2.** (**a**) Year-round (2000-2019) Week 3 $\widehat{NTR}$ anomaly correlation skill between nearest grid point $\widehat{NTR}$ reforecasts anomalies (SSH + IBE + VLM) and West Coast NOAA gauge station $NTR$ anomalies. (**b**) Difference between $\widehat{NTR}$ skill and

*NTR* persistence skill, (**c**) difference between $\widehat{NTR}$ skill and (SSH + VLM)-only skill, and (**d**) difference between $\widehat{NTR}$ skill and linearly detrended $\widehat{NTR}$ skill.

### 3.2.1 West Coast

In general, $\widehat{NTR}$ skill is highest along the southern to central West Coast, often exceeding 0.5 for leads through Week 6 (Fig. 2a). With the exception of a few gauge stations, notably those far up the Columbia River, West Coast reforecasts are also more

skillful than persistence (Fig. 2b). Note that for many of the stations between central California and Oregon, IFS Week 1 skill is only better than persistence with the IBE-correction (cf. Figs. 2b and 2c). Removing the linear trend minimally impacts West Coast skill, primarily only for longer leads in California (Fig. 2d).

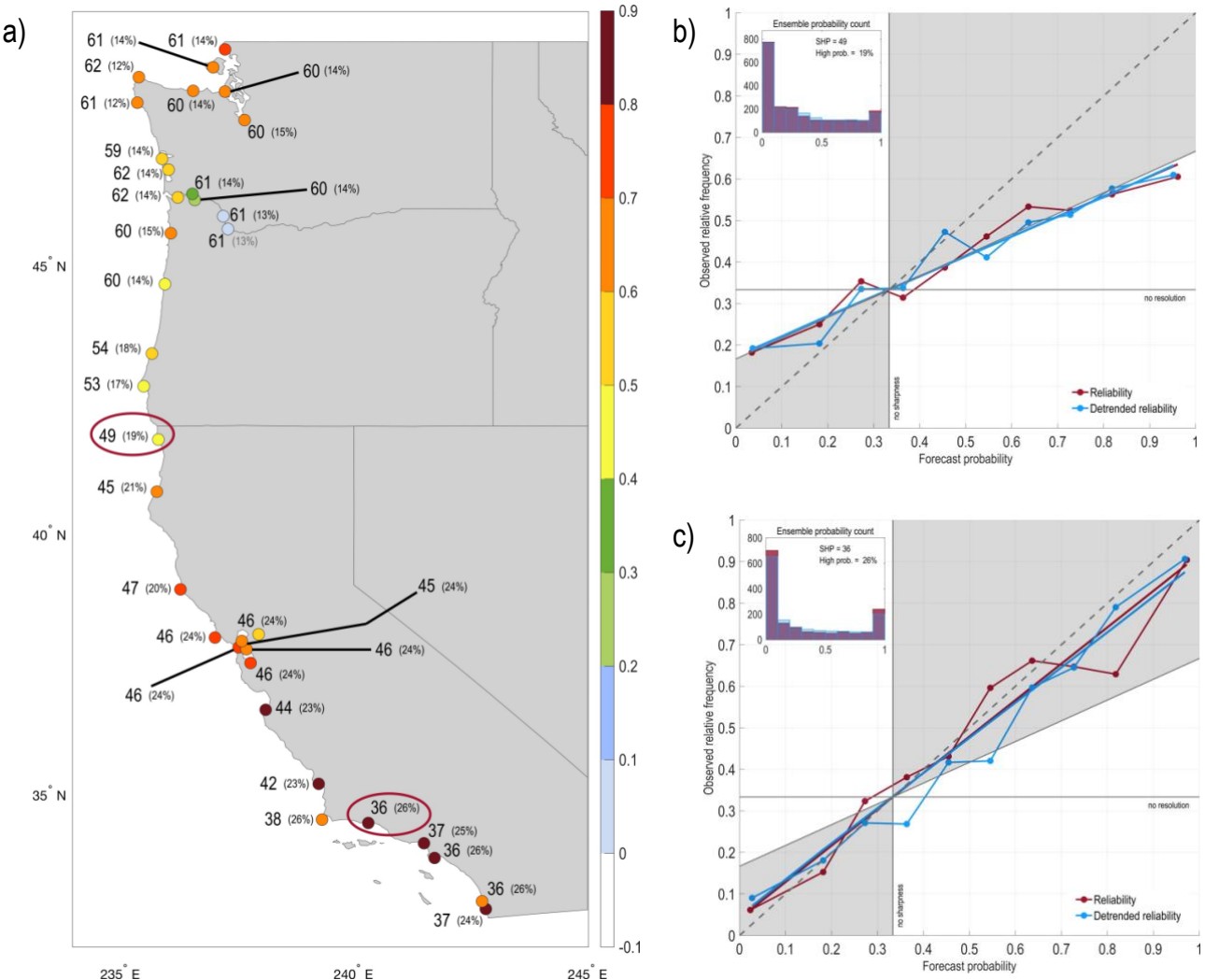

**Figure 3.** (**a**) Year-round (2000-2019) West Coast $\widehat{NTR}$ reliability (colored circles), sharpness (numbers), and percentage of forecasts that are in top forecast probability category versus lowest category (0-10% versus 90-100%). For reliability, slopes greater than 0.5 contribute to positive Brier skill scores; for sharpness, the scale spans 0-100, with smaller numbers representing sharper forecast distributions. Panels (**b**) and (**c**) show example reliability diagrams and sharpness distributions (ensemble probability counts) for two gauge stations, Crescent City, California and Santa Barbara, California, respectively, where the red ovals in panel (**a**) highlight the location of each station.

Reliability and sharpness along the West Coast tends to be largest in the south and decreases to the north (Fig. 3a). For example, reforecasts for gauge stations south of Arena Cove, CA have excellent reliability (regression slopes between 0.6-0.9) and sharpness (*SHP* between 30-50), with most gauge stations having roughly a quarter of their forecasts in the highest forecast probability category (e.g., Santa Barbara, CA in Fig. 3c). Even with the linear trend removed, most forecasts in central to southern California have excellent reliability and sharpness (e.g., Fig. 3c). Sharpness decreases roughly monotonically from south to north, but reliability has a minimum near the Oregon-California border (e.g., Crescent City, CA; Fig. 3b) with increasing values again from northern Oregon into Washington, apart from gauge stations extending well up the Columbia River. The IBE-correction increases reliability for all West Coast gauge stations, with regression slopes generally increasing by 0.2 slope units (not shown). The IBE-correction also mildly decreases sharpness (not shown); however, since sharpness for most gauge stations remains relatively high, this decrease is outweighed by the increased reliability realized with the IBE-correction. Linearly detrending the reforecasts has a relatively small impact on overall reliability and sharpness (not shown).

### 3.2.2 East Coast

In contrast to the West Coast, $\widehat{NTR}$ deterministic reforecast skill along the East Coast exceeds 0.5 only through Weeks 2-3 (Fig. 4a), though skill still nearly always exceeds that of persistence for all locations and at all leads (Fig. 4b). The IBE-correction improves skill for many East Coast gauge stations, particularly north of 35° N for Weeks 1 and 2 (Fig. 4c), so that $\widehat{NTR}$ skill exceeds persistence skill for the northernmost gauge stations for shorter lead times (cf. Figs. 4b and 4c). The linear trend also has a more significant impact on East Coast than West Coast skill, significantly contributing to Weeks 3-6, particularly for the Carolinas and Georgia (Fig. 4d). When both $\widehat{NTR}$ and persistence are linearly detrended, $\widehat{NTR}$ is still more skillful than persistence at all leads (not shown), consistent with the SSH results from Section 3.1.

With the exception of two gauge stations (Burlington, Delaware River, NJ and Bergen Point West Reach, NY), all of the $\widehat{NTR}$ reforecasts for the East Coast are at least minimally reliable (regression slopes >0.5), with some stations, particularly in the northeast, having quite high reliability (regression slopes >0.7). Moreover, for many of the northernmost and southernmost gauge stations, the forecasts also have reasonably sharp forecast probability distributions, with *SHP* values somewhere between 50-68 (Fig. 5). In contrast, most of the mid-Atlantic stations have *SHP* values >65 (that is, most reforecasts are near climatological probabilities), with only a small number of forecasts in the highest probability category.

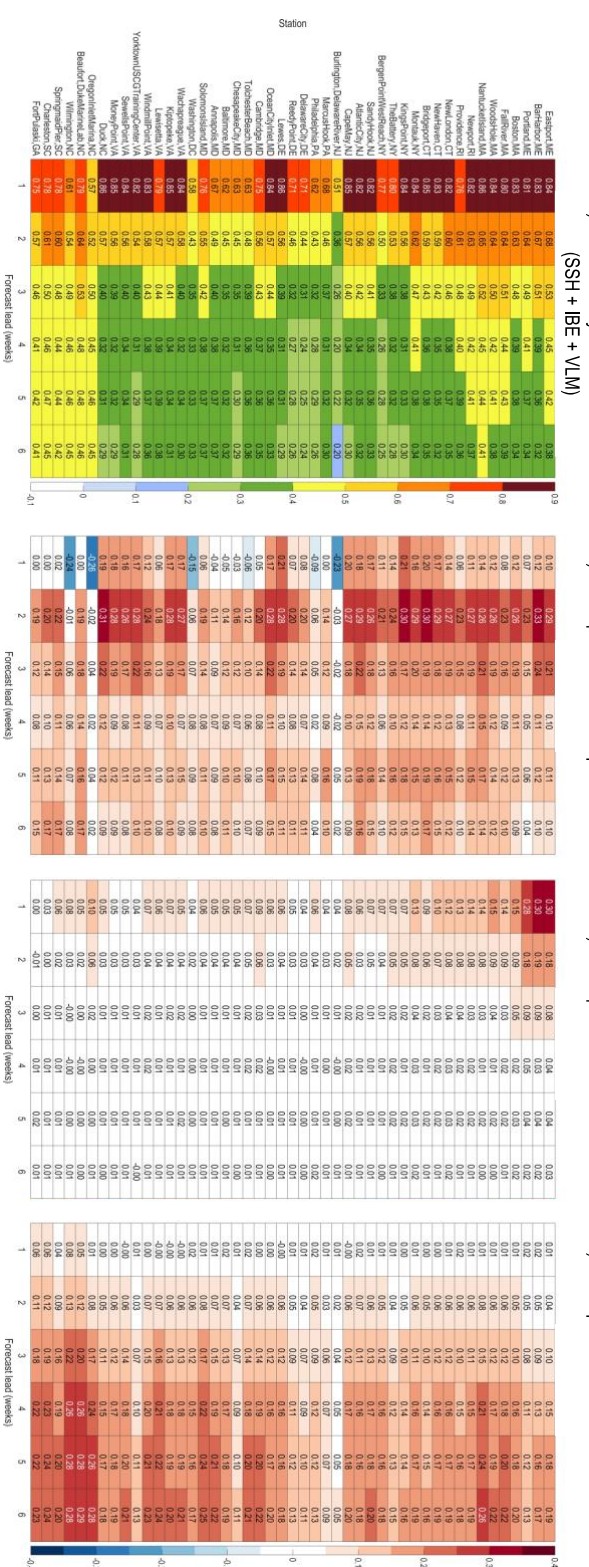

a) $\overline{NTR}$ anomaly correlation skill (SSH + IBE + VLM)

b) Skill improvement above persistence

c) Skill improvement due to IBE

d) Skill improvement due to linear trend

**Figure 4**. (**a**) Year-round (2000-2019) Week 3 $\widehat{NTR}$ anomaly correlation skill between nearest grid point $\widehat{NTR}$ reforecasts anomalies (SSH + IBE + VLM) and East Coast NOAA gauge station $NTR$ anomalies. (**b**) Difference between $\widehat{NTR}$ skill and $NTR$ persistence skill, (**c**) difference between $\widehat{NTR}$ skill and (SSH + VLM)-only skill, and (**d**) difference between $\widehat{NTR}$ skill and linearly detrended $\widehat{NTR}$ skill.

The IBE-correction impacts reliability differently depending on the region, improving it for gauge stations roughly north of New York City and decreasing it for the middle Atlantic Bight stations with no discernible change in sharpness (not shown), while there is little impact further south in the Carolinas. Thus, on balance, the notable benefits of the IBE-correction for the Northeast Coast appear to outweigh the small reduction in reliability for the middle Atlantic Bight region, particularly because the middle Atlantic Bight gauge stations have marginal sharpness that weighs against their overall usefulness. With the linear trend removed, reforecasts at all East Coast gauge stations become more overconfident and less sharp (e.g., Figs. 5b and 5c).

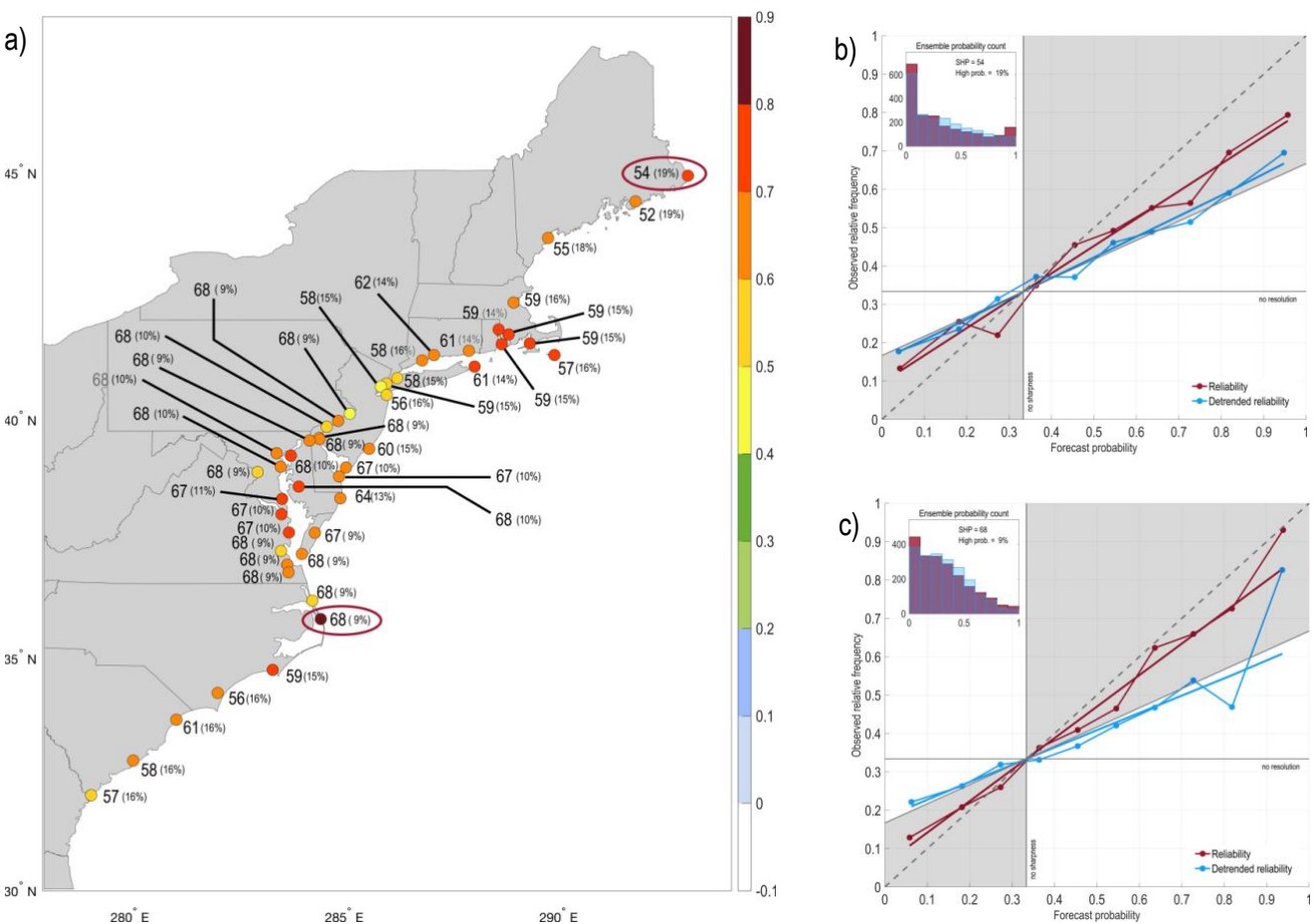

**Figure 5**. (**a**) Year-round (2000-2019) East Coast $\widehat{NTR}$ reliability (colored circles), sharpness (numbers), and percentage of forecasts that are in top forecast probability category versus lowest category (0-10% versus 90-100%). For reliability, slopes greater than 0.5 contribute to positive Brier skill scores; for sharpness, the scale spans 0-100, with smaller numbers representing sharper forecast distributions. Panels (**b**) and (**c**) show example reliability diagrams and sharpness distributions (ensemble probability counts) for two gauge stations, Eastport, Maine and Oregon Inlet Marina, North Carolina, respectively, where the

red ovals in panel (**a**) highlight the location of each station.

### 3.2.3 Gulf Coast

$\widehat{NTR}$ skill for many of the Gulf Coast stations remains at or above 0.5 until at least forecast Week 6 (Fig. 6a), and for nearly all gauge stations, the $\widehat{NTR}$ reforecasts are more skillful than persistence for Weeks 2-6 (Fig. 6b). The IBE modestly improves Weeks 1-2 reforecast skill for gauge stations between Mississippi and the southern tip of Florida (Fig. 6c), while the linear

trend greatly increases reforecast skill for all stations for leads at and beyond Week 2. While the steric and eustatic contributions to the linear trend are generally spatially uniform across the Gulf (e.g., Fig. 2.1 of Sweet et al. 2022), the effect of the VLM trend on reforecast skill is largely confined to gauge stations between Rockport, Texas and Dauphin Island, Alabama (Fig. 10a), where VLM improves correlation skill by roughly 0.1-0.2, particularly for forecast leads beyond Week 2 (Fig. S13c). Nevertheless, even when linear trend is removed from the reforecasts, reforecast skill (i.e., IFS-only SSH + IBE) exceeds

linearly detrended persistence skill for leads out to 2-3 weeks for Texas gauge stations and out to 6 weeks for stations between Louisiana and Florida (not shown).

Gulf Coast forecasts have good reliability (Fig. 7a), with regression slopes in the 0.6-0.9 range. There is one outlier gauge station relatively far inland (Berwick, Atchafalaya River, LA, Fig. 7b), with poorer reliability and with relatively few forecasts in the highest probability category (~10%). Most of the Gulf Coast stations have reasonably decent sharpness quite

similar to the St. Petersburg, FL gauge station (Fig. 7c). The IBE-correction increases reliability for gauge stations from Mississippi east to all of Florida (not shown), but appears not to impact reforecast skill in Louisiana or Texas. As for the East Coast, linearly detrending the reforecasts mildly decreases their reliability and sharpness (e.g., Fig. 7c).

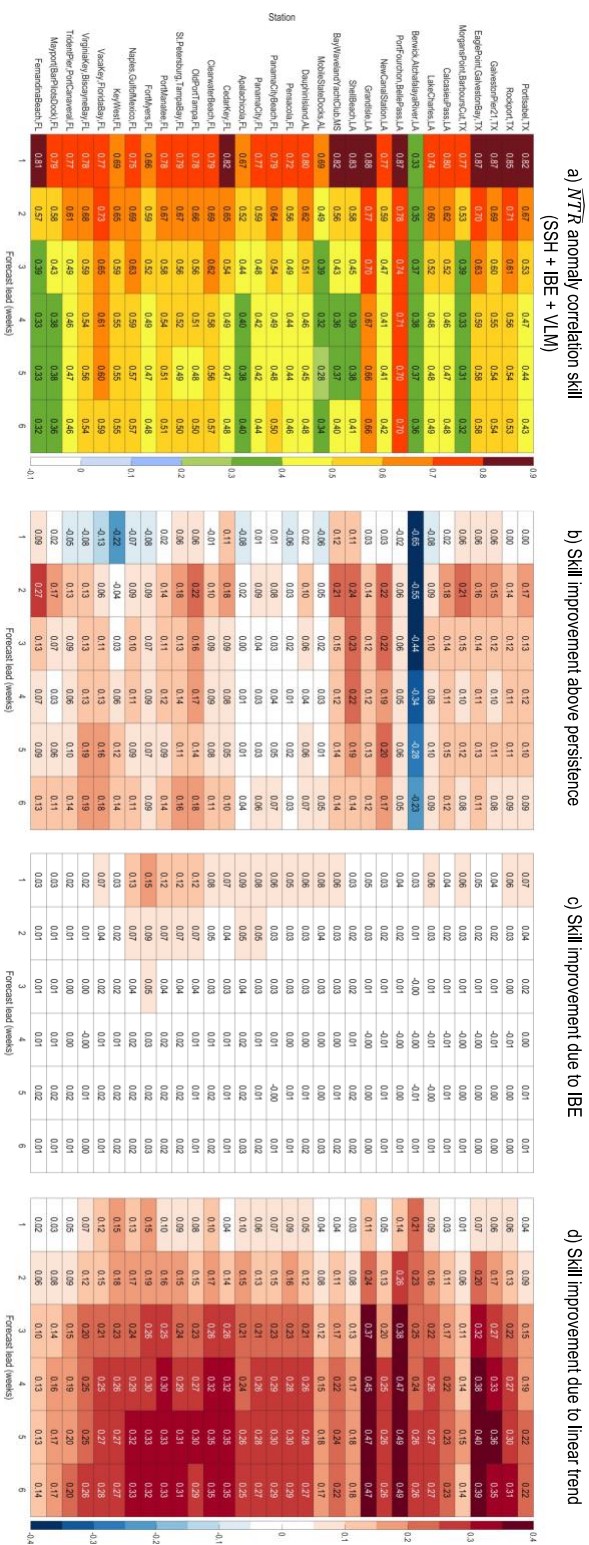

a) $\overline{NTR}$ anomaly correlation skill (SSH + IBE + VLM)

b) Skill improvement above persistence

c) Skill improvement due to IBE

d) Skill improvement due to linear trend

**Figure 6**. (**a**) Year-round (2000-2019) Week 3 $\widehat{NTR}$ anomaly correlation skill between nearest grid point $\widehat{NTR}$ reforecasts anomalies (SSH + IBE + VLM) and Gulf Coast NOAA gauge station $NTR$ anomalies. (**b**) Difference between $\widehat{NTR}$ skill and $NTR$ persistence skill, (**c**) difference between $\widehat{NTR}$ skill and (SSH + VLM)-only skill, and (**d**) difference between $\widehat{NTR}$ skill and linearly detrended $\widehat{NTR}$ skill.

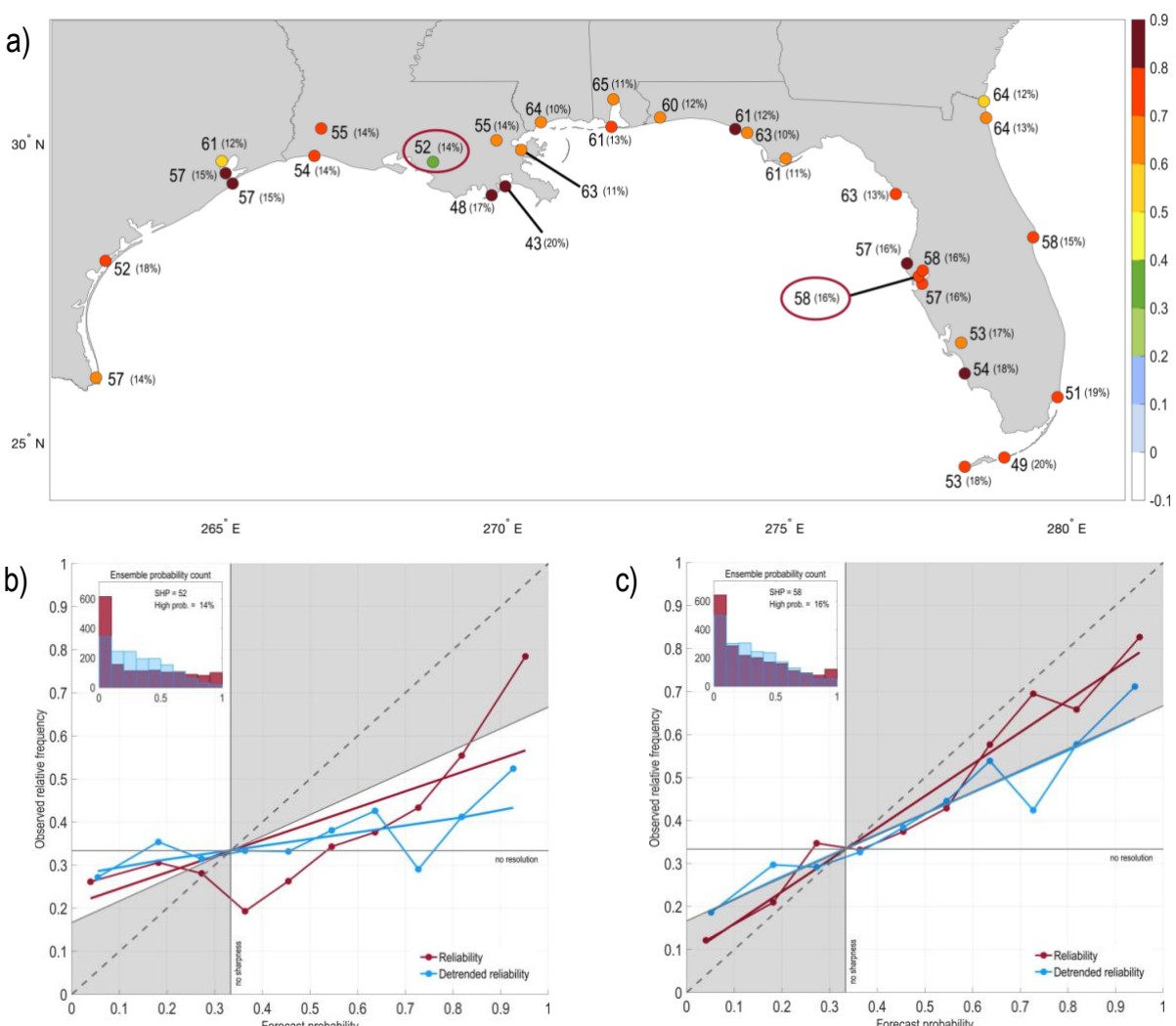

**Figure 7**. (**a**) Year-round (2000-2019) Gulf Coast $\widehat{NTR}$ reliability (colored circles), sharpness (numbers), and percentage of forecasts that are in top forecast probability category versus lowest category (0-10% versus 90-100%). For reliability, slopes greater than 0.5 contribute to positive Brier skill scores; for sharpness, the scale spans 0-100, with smaller numbers representing sharper forecast distributions. Panels (**b**) and (**c**) show example reliability diagrams and sharpness distributions (ensemble

probability counts) for two gauge stations, Berwick, Atchafalaya River, Louisiana and St. Petersburg, Florida , respectively, where the red ovals in panel **(a)** highlight the location of each station.

### 3.2.4 Alaska

Most Alaskan gauge stations have modest skill (anomaly correlation>0.5) through Week 3 (Fig. 8a), with skill exceeding persistence at all leads (Fig. 8b). The IBE notably contributes to reforecast skill for all stations out to two to three week lead times, and for a handful of stations, out to even longer leads (Fig. 8c). Reliability is also quite good, with all but a few stations having regression slopes >0.6 and moderate sharpness (Fig. 9). Except for Nome, Norton Sound, AK and Prudhoe Bay, AK, the IBE notably increases the slopes of the reliability regression lines (not shown), with many stations rising from poor reliability slopes (0.1-0.4) to quite useful slopes (0.5-0.8).

Removing the linear trend decreases overall skill for stations extending from roughly Kodiak Island southeast to Sitka (Fig. 8d). Along this section of the Alaskan coast, glacial isostatic rebound (see references in Section 2.2) is causing the land surface to rise, which is reflected in relatively large VLM rates over southeastern Alaska (Fig. 10a). Indeed, failing to account for VLM can lead to large errors in the $NTR$ anomaly time series; see, for example, the $\widehat{NTR}$ (blue lines) versus IFS-only (SSH + IBE) time series (orange lines) for Yakutat and Skagway, Alaska in Figs. 10b and 10c, respectively. In general, derived VLM rates (Kopp et al. 2014, Sweet et al. 2022) successfully account for a large portion of the negative linear $NTR$ trend, though the VLM corrected time series ($\widehat{NTR}$) at some gauge stations appear to more closely match the observed trend than for others (e.g., the VLM correction appears to underestimate the size of the land motion trend for Yakutat, Fig. 10b). Indeed, adjusting the IFS reforecasts with the predicted VLM rates does not completely resolve all trend issues, as evidenced by Port Alexander (Fig. 9c), where linearly detrending the reforecasts increases reliability. For stations in the Aleutian Islands and northwards to Prudhoe Bay, AK where isostatic rebound is either not occurring or is not significant, accounting for VLM only mildly increases reliability. Nevertheless, adjusting the IFS reforecasts to account for the VLM trend notably improves both deterministic and probabilistic $\widehat{NTR}$ reforecast skill between Kodiak Island and Sitka for all forecast leads, increasing anomaly correlations by 0.1-0.75 (Fig. S13a).

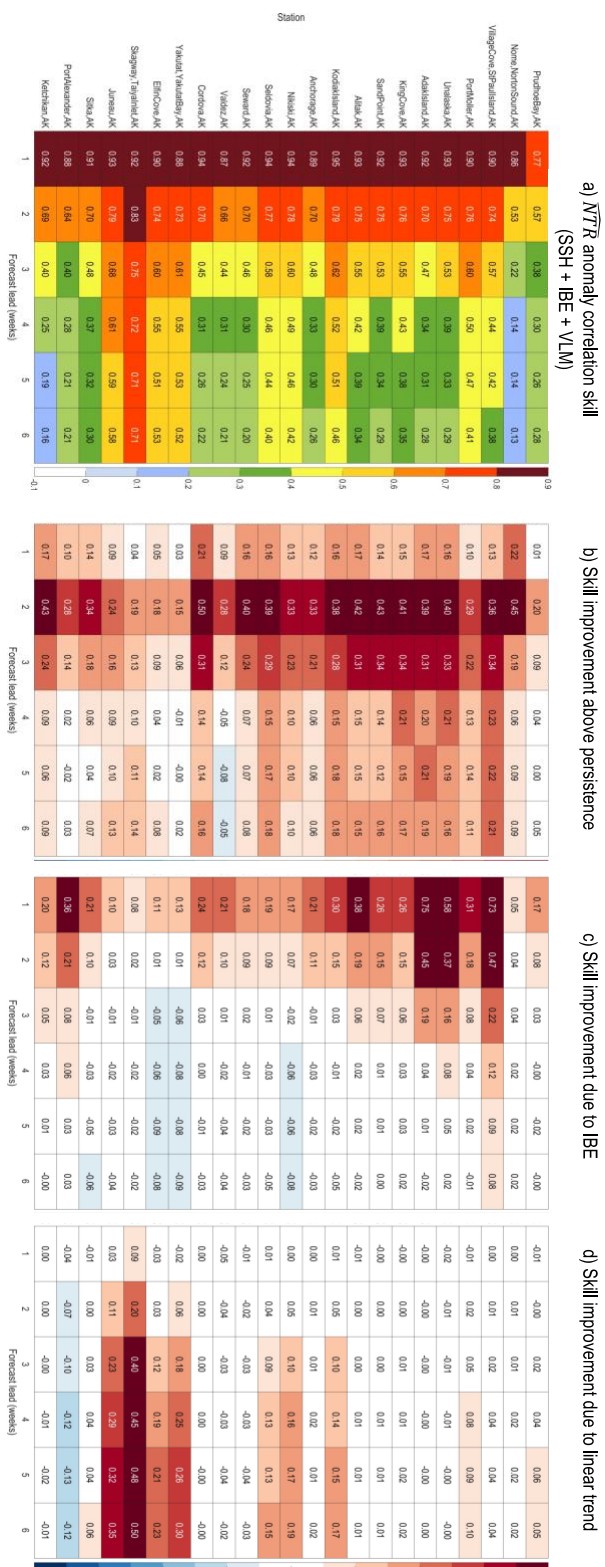

a) $\overline{NTR}$ anomaly correlation skill
(SSH + IBE + VLM)

b) Skill improvement above persistence

c) Skill improvement due to IBE

d) Skill improvement due to linear trend

**Figure 8**. (**a**) Year-round (2000-2019) Week 3 $\widehat{NTR}$ anomaly correlation skill between nearest grid point $\widehat{NTR}$ reforecasts anomalies (SSH + IBE + VLM) and Alaska NOAA gauge station $NTR$ anomalies. (**b**) Difference between $\widehat{NTR}$ skill and $NTR$ persistence skill, (**c**) difference between $\widehat{NTR}$ skill and (SSH + VLM)-only skill, and (**d**) difference between $\widehat{NTR}$ skill and linearly detrended $\widehat{NTR}$ skill.

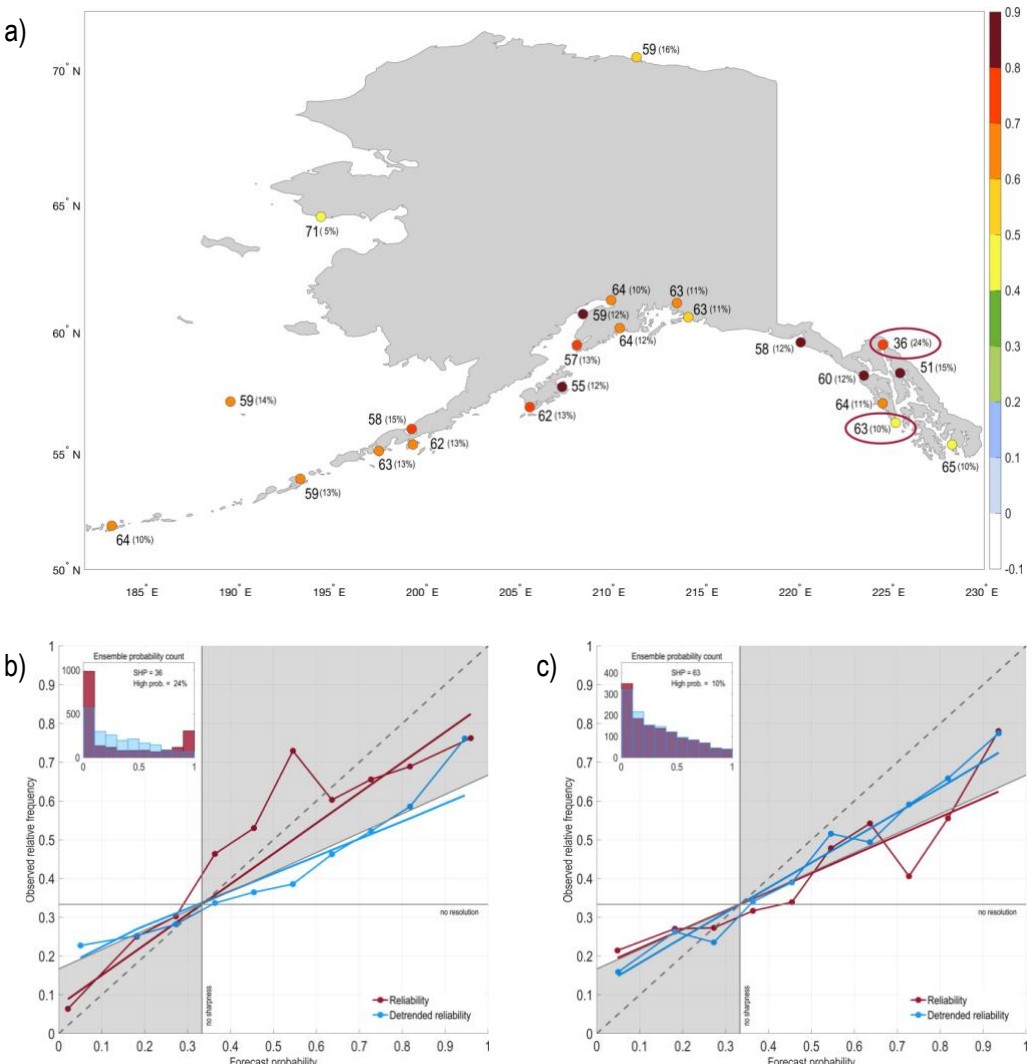

**Figure 9**. (**a**) Year-round (2000-2019) Alaska $\widehat{NTR}$ reliability (colored circles), sharpness (numbers), and percentage of forecasts that are in top forecast probability category versus lowest category (0-10% versus 90-100%). For reliability, slopes greater than 0.5 contribute to positive Brier skill scores; for sharpness, the scale spans 0-100, with smaller numbers representing sharper forecast distributions. Panels (**b**) and (**c**) show example reliability diagrams and sharpness distributions (ensemble probability counts) for two gauge stations, Skagway, Taiya Inlet, Alaska and Port Alexander, Alaska, respectively, where the red ovals in panel (**a**) highlight the location of each station.

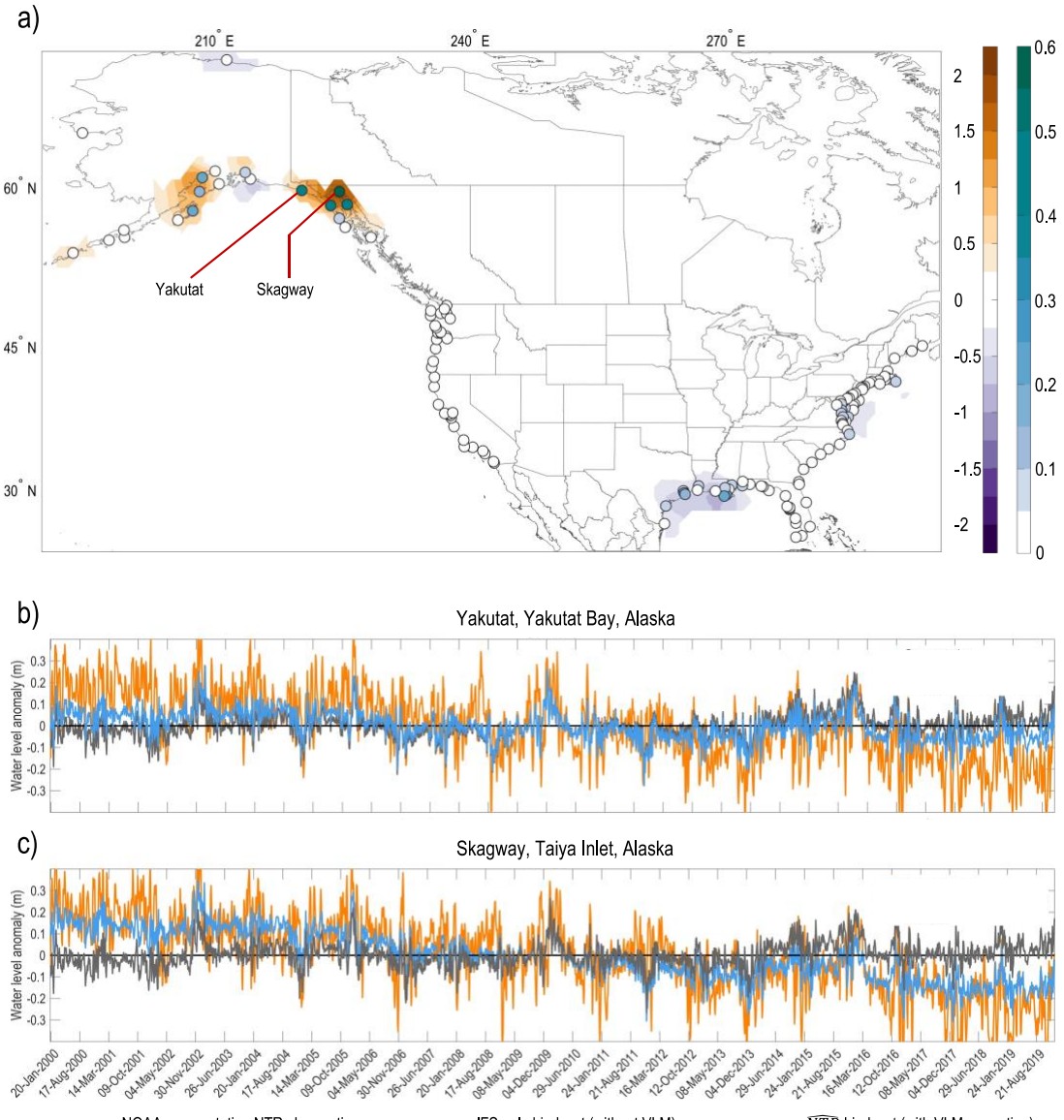

**Figure 10**. (**a**) Vertical land motion rates (cm/year) shown as colored contours with the NOAA gauge station locations used in the study shown as filled circular markers, where the color of the filled marker denotes the improvement in anomaly correlation skill when VLM is included in the reforecast, i.e., skill of $\widehat{NTR}$ reforecast minus skill of IFS-only reforecast (SSH + IBE). (**b**) Week 3 $\widehat{NTR}$ reforecast time series (SSH + IBE + VLM, blue line), IFS-only reforecast (SSH + IBE, gray line) and NOAA gauge station *NTR* time series (orange line) at Yakutat, Yakutat Bay, Alaska gauge station. (**c**) same as (**b**) but for the Skagway, Taiya Inlet, Alaska gauge station *NTR* time series (orange line) at Yakutat, Yakutat Bay, Alaska gauge station.

### 3.3 Seasonality of deterministic skill

While coastal flooding can happen in any season, all four United States sub-regions tend to have peaks in water levels exceeding the 90th percentile of the observed $NTR$ during the winter months (not shown). To understand how the IFS performs during these peak exceedance seasons, Figs. 11 and 12 group forecasts into four 3-month periods (JFM, AMJ, JAS, and OND), where for each season and gauge station, the latest forecast lead when $\widehat{NTR}$ anomaly correlation skill exceeds 0.5 is listed, both with the linear trend included (lefthand columns) and without (righthand columns).

With the linear trend included, forecasts skill for the East remains above 0.5 for forecast leads out to at least two weeks for all seasons and gauge stations, with the exception of the East Coast during late fall to early winter when skill only exceeds 0.5 for leads of one week (Fig. 11a). For the Gulf Coast, reforecast skill exceeds 0.5 for most stations through at least 2-3 weeks, but as far out as 6 weeks for many stations and seasons (Fig. 11c). With the linear trend removed, skill out to and beyond two week lead times is largely confined to late winter to early summer for the East Coast (Fig. 11b) and late fall to early spring for the Gulf Coast (Fig. 11d).

Reforecast skill is notably better for the central to southern portions of the West Coast, and is relatively insensitive to a linear trend (cf. Figs. 12a and 12b), with reforecast skill exceeding 0.5 through forecast Week 6 for nearly all California gauge stations throughout the year. For Oregon and Washington, skill exceeds 0.5 for forecast leads of 2-3 weeks. For Alaska, reforecast skill for most stations exceeds 0.5 through Week 3 for the cold season (Oct.-Mar.) but only through Week 2 for the warm season (Apr.-Sep.), with the exception of the stations between Yakutat and Juneau, where reforecast skill exceeds 0.5 at all lead times almost year-round (Figs. 12c and 12d).

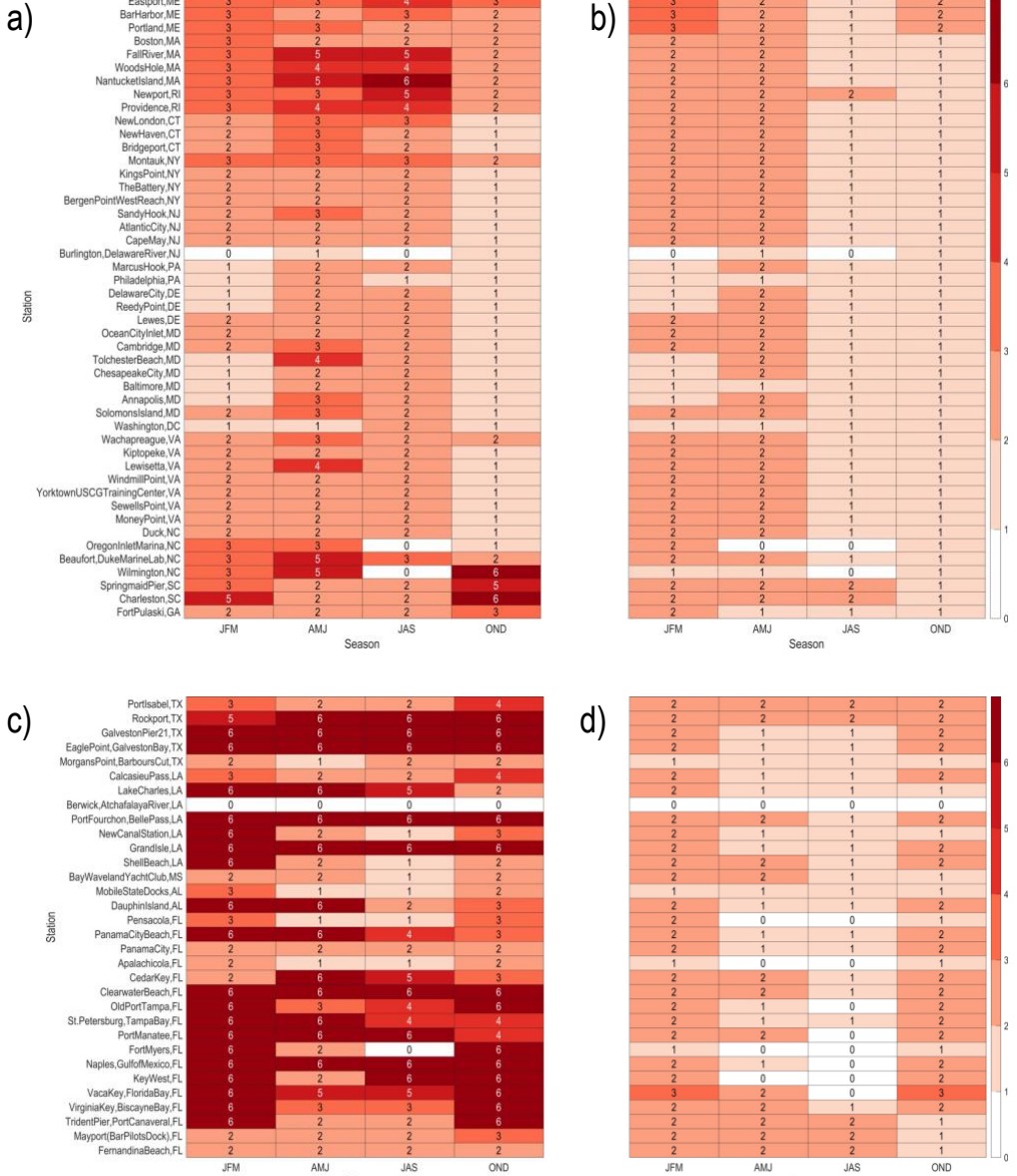

**Figure 11**. Seasonal cycle of $\widehat{NTR}$ reforecast skill (2000-2019) where the color bar (and number in each box) corresponds to the latest forecast lead (from week 1 to 6) when the anomaly correlation is greater than or equal to 0.5. In each panel, the seasonal cycle is split into four three-month periods, where the panels correspond to correlations between: (**a**) East Coast NOAA gauge station $NTR$ anomalies and $\widehat{NTR}$ reforecasts; (**b**) linearly detrended East Coast NOAA gauge station $NTR$ anomalies and linearly detrended $\widehat{NTR}$ reforecasts; (**c**) Gulf Coast NOAA gauge station $NTR$ anomalies and $\widehat{NTR}$ reforecasts; (**d**) linearly detrended Gulf Coast NOAA gauge station $NTR$ anomalies and linearly detrended $\widehat{NTR}$ reforecasts.

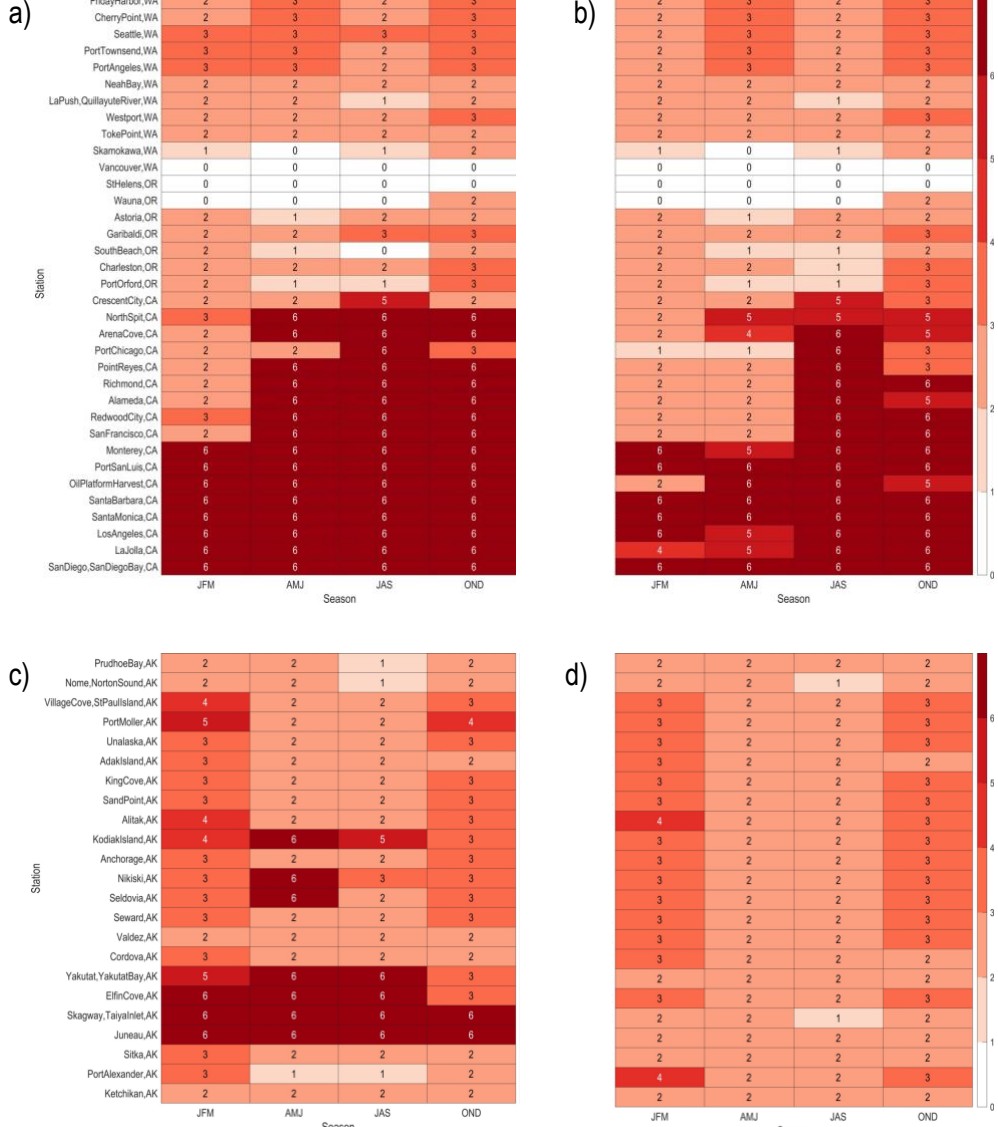

**Figure 12**. Seasonal cycle of $\widehat{NTR}$ reforecast skill (2000-2019) where the color bar (and number in each box) corresponds to the latest forecast lead (from week 1 to 6) when the anomaly correlation is greater than or equal to 0.5. In each panel, the seasonal cycle is split into four three-month periods, where the panels correspond to correlations between: (**a**) West Coast NOAA gauge station $NTR$ anomalies and $\widehat{NTR}$ reforecasts; (**b**) linearly detrended West Coast NOAA gauge station $NTR$ anomalies and linearly detrended $\widehat{NTR}$ reforecasts; (**c**) Alaska NOAA gauge station $NTR$ anomalies and $\widehat{NTR}$ reforecasts; (**d**) linearly detrended Alaska NOAA gauge station $NTR$ anomalies and linearly detrended $\widehat{NTR}$ reforecasts.

## 4 Conclusions

The primary goal of this paper is to assess the suitability of the current generation of forecast models for making coastal inundation forecasts on subseasonal timescales. Indeed, we find that the deterministic skill of the IFS and CNRM exceeds that of damped persistence for many US coastal regions for forecast leads extending out to 2-3 weeks, with the IFS continuing to have skill above damped persistence for longer leads through Week 6. However, when reforecasts and persistence reforecasts are linearly detrended, the skill of both models over persistence increases, highlighting the ability of the models to successfully simulate water level anomalies related to subseasonal climate variability. When a simple bias correction is applied to the IFS' probability distribution, the IFS generally has 'useful' reliability (Weisheimer and Palmer, 2014) that contributes positively to the Brier skill score (Mason, 2004), though the need for the probability bias correction highlights that the model ensemble spread is consistently underdispersive. Thus, our results suggest that the current generation of operational forecast models provide predictions of coastal inundation with sufficient skill to form the basis for improved forecast guidance of high tide flooding predictions on subseasonal timescales.

We have also demonstrated the regional and forecast lead time dependence of weekly $\widehat{NTR}$ prediction skill. California has by far highest skill, with anomaly correlation skill above 0.6 through at least Week 6. Many portions of Alaska also have anomaly correlation skill that reaches or exceeds 0.5 for leads through Week 3, with some stations in southeastern Alaska having skill exceeding 0.5 through Week 6 once past VLM trends are taken into account. On the East Coast, on the other hand, skill is quite low for the mid-Atlantic states but is relatively higher south of Cape Hatteras and further north along the New England coast (Figs. 1 and 4), where anomaly correlation skill generally ranges from 0.4 to 0.5 for forecast leads out to at least Week 3. For many Gulf Coast stations, skill remains above 0.4 to 0.5 for leads out to at least Week 6. However, for many Alaskan and East and Gulf Coast gauge stations, a large fraction of the reforecast skill can be attributed to the linear $\widehat{NTR}$ trend, which includes steric and eustatic trends that are explicitly accounted for in the model reforecasts, as well as VLM trends that are added via post-processing.

How the predicted IBE impacts $\widehat{NTR}$ reforecast skill also appears to strongly depend upon forecast lead time and region. Accounting for the IBE primarily improves reforecast skill during forecast Weeks 1 and 2, which is consistent with predictable IBE signals being limited to weather timescale atmospheric pressure fluctuations (e.g., the timescales suggested in Fig. 2 of Woodworth et al. 2019). Still, the IBE is essential since, for many gauge stations, Week 1 IFS reforecasts only have more skill than persistence when the IBE-correction is applied. The regional dependence of IBE-related skill improvement is at least somewhat consistent with the idea that the IBE is more important at higher latitudes (Chelton and Davis, 1982), particularly in the Gulf of Alaska and the northern portion of the East Coast (Wunsch and Stammer, 1997; Ponte, 2006). However, it is likely that the geographical dependence of MSLP reforecast skill itself also plays a role. For example, the IFS at Weeks 3-6 tends to have higher MSLP skill for the East Coast and the Aleutian Islands than for the West Coast and mainland Alaska (Albers and Newman, 2019, see their Figs. 1 and S2). Likewise, while the importance of the IBE seems to monotonically increase from south to north along the East Coast (Figure 4), the middle of the West Coast appears to benefit

more from including the IBE than do the northernmost West Coast gauge stations (Fig. 2). Overall, the skill improvement we find with a postprocessed IBE correction suggests that future forecast model development would benefit from the explicit inclusion of the IBE.

The relatively low $\widehat{NTR}$ reforecast skill for many regions suggests that usable $NTR$ forecast guidance may benefit from identifying 'forecasts of opportunity'; that is, when predictions are *expected* to have skill at the time of forecast issuance (e.g., Albers and Newman, 2019; Lang et al. 2020; Mariotti et al. 2020; and references therein). For example, when the MJO, ENSO, or the stratosphere may produce predictable oceanic waves and/or teleconnections in sea level pressure and surface wind (Barnston et al., 2019; DelSole et al., 2017; Kim et al., 2018; Tripathi et al., 2015; Vitart & Molteni, 2010, Albers and Newman, 2021) may also be times when coastal $\widehat{NTR}$ anomalies are particularly predictable, which may be helpful for issuing more definitive guidance to many coastal communities and stakeholders. For the southern and central portion of California, the relatively high reforecast skill likely can be anticipated as a consequence of coastal Kelvin waves and remote wind variability from coupled modes of variability including ENSO (Menéndez and Woodworth, 2010; Arcodia et al., 2024, Amaya et al. 2022). For the East Coast, identifying predictable Gulf Stream subseasonal anomalies may lead to identifying times of higher $\widehat{NTR}$ reforecast skill from Florida through the Gulf Stream separation point near Cape Hatteras, since Gulf Stream variability and coastal SLAs are known to be correlated (Ezer et al., 2013; Ezer, 2016; Chi et al., 2023). In contrast, the increase in coastal reforecast skill further north along the New England portion of the coast may be associated with the NAO, whose skill on subseasonal timescales can be anticipated from prior stratospheric and tropical conditions (e.g., Albers et al. 2021 and references therein), since the NAO has a stronger influence further north in New England versus the mid-Atlantic (Hurrell et al. 2003, Visbeck et al. 2003). Our study will form the basis for future work that uses dynamical models to better predict these subseasonal forecasts of opportunity.

*Data availability.* The data used in this study are available from the following locations: GLORYS https://data.marine.copernicus.eu/product/GLOBAL_MULTIYEAR_PHY_001_030/description ECMWF IFS and CNRM reforecasts https://apps.ecmwf.int/datasets/data/s2s/levtype=sfc/type=cf/ VLM (https://oceanservice.noaa.gov/hazards/sealevelrise/Sea_Level_Rise_Datasets_2022.zip); and NOAA tide gauges (https://api.tidesandcurrents.noaa.gov/mdapi/prod/ ).

*Author contributions.* Conceptualization, project administration, and supervision by JRA and MN. Data curation by JRA and YW. Formal analysis, investigation, methodology, visualization, and writing of original draft by JRA. Project administration by JRA and MN. Writing, review and editing by JRA, MN, MAB, WS, TX, and YW.

*Competing interests.* The authors declare that they have no conflict of interest.

*Acknowledgements.* The authors would like to thank Drs. Matthew Widlansky, Chia-Wei Hsu, Dillon Amaya, and Paige Hovenga for discussions that improved the manuscript. JRA, MN, BS, YW, TX acknowledge the support from NOAA cooperative agreement NA22OAR4320151 and the U.S. DoC/NOAA/Bipartisan Infrastructure Law (BIL).

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
