# Peer review of "Assessing Subseasonal Forecast Skill for Use in Predicting US Coastal Inundation Risk"

_EGUsphere, 2025_

## Referee Comment (RC2)

I appreciate the opportunity to review *"Assessing subseasonal forecast skill for use in predicting US coastal inundation risk"*. This is a well-needed study, which I think merits speedy publication. I only have minor revisions to recommend considering (* deserving more attention), as detailed below (since the manuscript numbering is every 5th line, numbers are approximate).

Also, please accept my apology for tardiness in submitting this review. While I might be slower than the ideal reviewer, I do wonder if more attention to the availability of much speedier AI reviewers is warranted. For the authors' and editor's consideration, I've provided below my review a copy of the review I sought from ChatGPT o3.

L25 "trillions of dollars in property…" is a bit unclear.
L45 Could also cite Long et al. 2021. Specifically, the finding of no seasonal skill for the U.S. East Coast.
L70 Also consider Widlansky et al. 2017, which introduced the approach for operational seasonal sea level forecast products in the tropical Pacific islands (i.e., high-tide outlook -- = tide prediction + sea level trend + sea level anomaly forecast).
L70 "along shallower coasts" Do you mean "coasts along shallower shelves"?
L75 Abbreviate ECMWF
L75 Re: role of altimetry assimilation on forecasting skill, consider Widlansky et al. 2023.
L90 Abbreviate CNRM
L100 Feng et al. 2024 assessed a number of ocean reanalyses for U.S. Gulf and East Coast monthly sea levels.
L110 Method of separating into three components the HTF outlook was originally developed by Widlansky et al. 2017 (for tropical Pacific Islands).
L110 Clarify a bit about how the trend contribution is assessed (I found the actual assessment sufficient, just not its description).
L120 "data is scored and available" a bit unclear.
L130 IB effect in the subheading could be abbreviated.
L130 "remove static" and dynamic atmospheric pressure effects (i.e., the Dynamic Atmospheric Correction applied to altimetry measurements.
L130 The explanation why forecasts do not include the IB-effect is incomplete. The altimetry assimilation does not include the IB-effect AND models have the IB-effect turned off.
L140 Missing subscript in the density unit.
L140 Consider Feng et al. 2024 again here.

*L150 The VLM correction aspect of this study is overly complex, and perhaps worse than taking a much simpler approach. VLM spatial variability is substantial on sub 1 deg resolution. Why not approximate VLM as the difference between the altimetry (or GLORYS12) and tide gauge observation?

L165 "respectively" unclear

L205 "near the Bahamas" IFS seems more skillful there.

*L220 Figure 1 should also show the damped-persistence skill in map form.

L355 Consider the IB-effect assessment presented in Long et al. 2021

*L385 Figure 10b is an example of why a better (simpler) approach to the VLM correction is warranted.

L455 Again, consider the IB-effect assessment presented in Long et al. 2021

L480 The last paragraph seems a bit incomplete.

*Prompt: Please review this manuscript.*

*https://egusphere.copernicus.org/preprints/2025/egusphere-2025-897/egusphere-2025-897.pdf*

ChatGPT o3 review

**Overall evaluation**

This manuscript tackles a timely, clearly-defined problem: *can current subseasonal-to-seasonal (S2S) forecast systems add skill – beyond damped persistence – to operational high-tide-flood (HTF) outlooks for U.S. tide-gauge sites?* Using a 20-year hindcast set (2000-2019) from two global coupled models (ECMWF-IFS and CNRM) that assimilate altimetry, the authors compute deterministic (ACC, RMSE) and probabilistic (ROCSS, RPS) skill for non-tidal residual (NTR) anomalies at 80+ gauges, test simple post-processing steps (inverse-barometer correction and vertical-land-motion adjustment) and compare skill to damped persistence. They find that IFS beats damped persistence out to 5–6 weeks and that modest post-processing gives further gains at short leads. Results are positioned as evidence that NOAA's monthly HTF product could incorporate model guidance today. The study is well organised, methodologically transparent, and squarely within the scope of *EGU Ocean Science*. I recommend **minor revision** before publication.
* * *
**1. Major comments**

| # | Comment | Recommendation |
|---|---------|----------------|
| **M1** | **Justification of model choice.** The manuscript asserts that IFS and CNRM are chosen because they assimilate altimetry and predict global-mean sea level, "whose absence in other systems complicates assessment" EGUsphere. But several S2S systems (e.g., NCEP GEFSv12, JMA CPS2, BoM ACCESS-S2) now include dynamic sea-level components. | Add a short table (or SI) listing major S2S models and clarifying why each was or was not analysed. This will strengthen the argument that the two systems are genuinely state-of-the-art and representative. |
| **M2** | **Downscaling / coastal processes.** Using the nearest ocean grid point ignores shelf dynamics, bathymetry, and local surge processes acknowledged by the authors (lines 60–67) EGUsphere. While a full dynamical downscaling is beyond scope, the implications deserve deeper discussion. | Include a sensitivity test (perhaps for two contrastive regions) using an along-shelf average or a simple statistical bias-corrector to demonstrate robustness, or else clarify that such work is left to future studies. |
| **M3** | **Vertical land motion (VLM) adjustment.** The method for VLM correction is briefly described but key details are missing: data sources (GPS? GIA models?), temporal resolution, and whether tectonic/anthropogenic motions are separated. | Provide a concise paragraph in Methods plus a reference to underlying datasets or provide the VLM time-series in the supplement. |
| **M4** | **Forecast ensemble size and handling of lead-time drift.** The skill metrics are computed on ensemble means, but the manuscript does not state how many ensemble members each hindcast system provides and whether mean bias drifts with lead. | Report ensemble sizes and show lead-dependent mean bias (could be a small inset figure). Specify whether bias correction is applied before or after computing skill. |
| **M5** | **Practical guidance to HTF operations.** The conclusions state that "current forecast systems have sufficiently high skill to be used in support of operational coastal flood guidance" EGUsphere, yet no example decision metric is shown. | Add one applied example (e.g., conditional flood probability at Charleston for a selected event) to illustrate how the models would change an existing NOAA HTF outlook. |

**2. Minor / technical comments**

1. **Terminology consistency** – sometimes "non-tidal residual", elsewhere "sea-level anomaly". Standardise to NTR (define once).

2. **Figure 2 colour-bar** – red/blue diverging palette is illegible for colour-blind readers; consider a perceptually-uniform scheme.

3. **Lines 115–120** – cite the version numbers of IFS (cycle 47r3?) and CNRM (CM6-1).

4. **Equation 3** – clarify whether trend term is removed before ACC computation or only from forecasts.

5. **ROCSS significance** – briefly state the bootstrap or Monte-Carlo procedure used to assess the 95 % confidence shown.

6. **References** – duplicate DOI appears for Long et al. 2021 and "submitted" paper (line 590) [EGUsphere](); check.

7. **Typos** – "Balmeseda" → "Balmaseda" several times; "probabilistic skill *scores*" not "score" in abstract.
* * *
**3. Strengths**

- Addresses an operationally important gap (2–6 week lead flood guidance).

- Uses long, consistent hindcast sets and multiple complementary skill metrics.

- Post-processing steps are simple and reproducible, thus attractive to agencies.

- Manuscript is generally well written and logically structured.

**4. Summary recommendation**

With the clarifications suggested above – particularly the added context on model selection, VLM methodology, and an applied forecast example – the paper will provide a convincing, actionable assessment of subseasonal coastal-sea-level predictability and be a valuable reference for both research and operations.

---

## Author Response (AR1)

*Response to Reviewer #1's Comments on:*
*"**Assessing Subseasonal Forecast Skill for Use in Predicting US Coastal Inundation Risk**"*
by Albers et al. (EGU Ocean Science -
https://doi.org/10.5194/egusphere-2025-897

We thank Reviewer #1 for their thoughtful comments and suggestions, which we address below.

Reviewer wrote: *Page 2 Line 50 Sweet and Zervas 2011 is not in the references.*

Our response: Reference added.

Reviewer wrote (two reviewer comments are addressed here):
*Page 3 can you explain the forecast models a little more. For example what are their time resolution, how many forecast steps are there or how far ahead do they predict. Later on we see that its up to six weeks.*

*…and…*

*Line 113*
*You say that "All reforecast and verification datasets are determined using seven day running mean anomalies" This to me is not very clear. What is you basic time step in this. The water level data is generally 6 minute or 1 hour or have you gone to daily values? Are you taking these values ad producing a seven day running mean or are you removing a seven-day running mean?*

Our response: We agree that this section should be more clear. We have added a fair amount of additional text (lines 116-130) explaining that the models are available as daily average data and that the IFS is run out to 46 days and the CNRM is run out to 46 days, as well as more completely defining how the climatologies of each forecast and verification dataset are created. We also added text on lines 23-24 of the Supplement that clarify how the IFS climatology is created.

Reviewer wrote: *Line 114. What do you exactly mean by 20- or 25- year climatologies? You say in thesupplement the first four harmonics (plus the mean). So I would assume that would mean Sa and Ssa but what is generally the other two?*

Our response: This method is commonly used (e.g., Epstein J. Clim. 1988) to remove a smooth approximation to the seasonal cycle when only limited data is available (e.g., 20 years like in this paper). For example, if we had 1000 years of data, then taking the 1000-year average of each calendar day would produce a very nice smooth 365-day approximation to the seasonal cycle that could be used as a climatology for creating deseasonalized anomalies. However, if that same dataset was truncated to 20 years, the bulk features of the climatology would look the same, but it would be much noisier because of sampling issues. Doing a Fourier decomposition of the noisy climatology and then keeping the first handful of harmonics (plus the mean) approximates the smooth seasonal cycle quite well. The number of harmonics to keep is in some sense arbitrary, but in practice, only including the first 1-3 typically does not keep enough variance to approximate the total pattern (this is pretty clear if you plot say the sum of the first 2-3 harmonics and compare then to the non-Fourier-truncated plot). In principle one could include more than four harmonics, but then the pattern starts to look noisy again. In our experience, including the first three harmonics is often not enough, but including more than four achieves only diminishing returns. That said, our results are not sensitive to keeping more than 4 harmonics (I have in the past tested out to 12 and not found much of a difference), so keeping the first four harmonics tends to be a pretty good balance.

That all said, your point is well-taken that a reference is useful, so we have added the Epstein (1988) reference to the paper on line 124.

Reviewer wrote: *Page 9, Line 151. Sweet et al 2022 seems to be missing in the references*

Our response: Reference added.

Reviewer wrote: *Page 10. Equation 2. You talk about IBE and VLM corrected but I think you should be clearer on what you mean. Are you adding them on to one thing or removing them from another. I presume you are adding on to the forecasts to make the predicted NTR as close to the observed NTR as possible.*

Our response: We have clarified this on lines 169-173.

Reviewer wrote: *Page 16. Line 278. I don't know "increase from north to south" seems better than "decrease from south to north"! Or even just "increase towards the south" or "decrease to the north"*

Our response: Fixed.

*Response to Reviewer #2's Comments on:*
**"*Assessing Subseasonal Forecast Skill for Use in Predicting US Coastal Inundation Risk*"**
by Albers et al. (EGU Ocean Science -
https://doi.org/10.5194/egusphere-2025-897

We thank Reviewer #2 for their helpful comments and suggested references, which improved the manuscript. We address the Reviewer's concerns in detail below.

Reviewer wrote: *L25 "trillions of dollars in property…" is a bit unclear.*

Our response: Agreed, that sentence should be more clear, we have edited it.

Reviewer wrote: *L45 Could also cite Long et al. 2021. Specifically, the finding of no seasonal skill for the U.S. East Coast.*

Our response: This sentence discusses subseasonal time scales, while Long et al. 2021 focuses on seasonal. However, we have included the Long et al. (2024) reference at several other points in the paper (line 77 and 248).

Reviewer wrote: *L70 Also consider Widlansky et al. 2017, which introduced the approach for operational seasonal sea level forecast products in the tropical Pacific islands (i.e., high-tide outlook -- = tide prediction + sea level trend + sea level anomaly forecast).*

Our response: Thank you for bringing this reference to our attention. We assume you were referring to line 33 where the Dusek et al. reference is cited for this decomposition rather than line 70; we have included it there.

Reviewer wrote: *L70 "along shallower coasts" Do you mean "coasts along shallower shelves"?*

Our response: We have clarified this sentence.

Reviewer wrote: *L75 Abbreviate ECMWF*

Our response: The ECMWF is only mentioned twice in the main body of the paper, so we prefer not to introduce an additional acronym (we instead utilize the associated 'IFS' acronym when referring to the ECMWF model).

Reviewer wrote: *L75 Re: role of altimetry assimilation on forecasting skill, consider Widlansky et al. 2023.*

Our response: Reference added.

Reviewer wrote: *L90 Abbreviate CNRM*

Our response: We prefer to write out the name of the forecast center in full here.

Reviewer wrote: *L100 Feng et al. 2024 assessed a number of ocean reanalyses for U.S. Gulf and East Coast monthly sea levels.*

Our response: Reference added.

Reviewer wrote: *L110 Method of separating into three components the HTF outlook was originally developed by Widlansky et al. 2017 (for tropical Pacific Islands).*

Our response: Here we are specifically referring to the NOAA/CO-OPS operational HTF, so the Widlansky et al. 2017 reference does not fit. We did insert the Widlansky et al. 2017 reference again on lines 33 and 248.

Reviewer wrote: *L110 Clarify a bit about how the trend contribution is assessed (I found the actual assessment sufficient, just not its description).*

Our response: We have added a clarifying sentence on lines 113-115.

Reviewer wrote: *L120 "data is scored and available" a bit unclear.*

Our response: We have clarified this sentence.

Reviewer wrote: *L130 IB effect in the subheading could be abbreviated.*

Our response: We prefer not to abbreviate in the subheading.

Reviewer wrote: *L130 "remove static" and dynamic atmospheric pressure effects (i.e., the Dynamic Atmospheric Correction applied to altimetry measurements.*

Our response: Thank you for noticing that, we have edited the text on lines 137-138.

Reviewer wrote: *L130 The explanation why forecasts do not include the IB-effect is incomplete. The altimetry assimilation does not include the IB-effect AND models have the IB-effect turned off.*

Our response: We have added additional clarifying text on lines 140-141.

Reviewer wrote: *L140 Missing subscript in the density unit.*

Our response: We do not see any missing subscript in the density unit, however we did notice that we refer to the IBE in equation (1) on line 152, but we should have referred to $\eta_{ibe}$, which we have fixed.

Reviewer wrote: *L140 Consider Feng et al. 2024 again here*

Our response: Reference added.

Reviewer wrote: *\*L150 The VLM correction aspect of this study is overly complex, and perhaps worse than taking a much simpler approach. VLM spatial variability is substantial on sub 1 deg resolution. Why not approximate VLM as the difference between the altimetry (or GLORYS12) and tide gauge observation?*

Our response:

The determination of VLM is complex, and there are several different estimates we could have used, which we now note on lines 160-161). We chose a standard approach based on VLM rates that have been vetted and are published. Deriving our own rates, which would not have been previously published, is beyond the scope of this work.

Reviewer wrote: *L165 "respectively" unclear*

Our response: Respectively refers to the 'reforecasts' and 'verifications'; because these are the two nouns immediately preceding 'respectively', we think this is clear.

Reviewer wrote: *L205 "near the Bahamas" IFS seems more skillful there.*

Our response: Indeed, we should have been more precise with this part of the statement, we have changed it to 'northeast of the Bahamas'; the text has been changed to reflect this.

Reviewer wrote: *L220 Figure 1 should also show the damped-persistence skill in map form.*

Our response: The focus of this paper is on skill at NOAA coastal gauge stations, which is why we show damped persistence skill for all coastal gauge stations at all forecast leads. Figure 1 was included to (1) provide a comparison of prediction skill between the IFS and CNRM models; (2) orient the reader regarding station locations; and (3) show that near-shore ocean SSH skill tends to be spatially correlated with gauge station skill. However, given that the focus

of the manuscript is not open ocean skill, and the fact that the manuscript already has 12 main manuscript figures and 13 supplemental figures, we would prefer not to add additional figures.

Reviewer wrote: *L355 Consider the IB-effect assessment presented in Long et al. 2021*

Our response: Long et al. 2021 assessed the IB-effect by removing it from the data. We respectfully disagree with this approach for our purposes here, since then we are not verifying the quantity that we actually want to predict (i.e., the total NTR). Instead, we have taken the very deliberate approach of assessing the best possible skill of tide gauge NTR using the IFS forecast system. This is the point of the discussion related to eqn. 2, which we have expanded upon in the revision.

Reviewer wrote: *L385 Figure 10b is an example of why a better (simpler) approach to the VLM correction is warranted.*

Our response: Again, it is not obvious to us why using published VLM rates is somehow more complex than deriving our own unpublished VLM estimates (which could be an entire paper on its own).

Reviewer wrote: *L455 Again, consider the IB-effect assessment presented in Long et al. 2021*

Our response: See previous response on this topic.

Reviewer wrote: *L480 The last paragraph seems a bit incomplete.*

Our response: This paragraph has been revised and we have added a final sentence to indicate future work here.

Regarding the ChatGPT portion of this review: We have some ethical concerns about uploading an unpublished manuscript into ChatGPT without the authors' permission. Relatedly, the use of ChatGPT for peer review is inappropriate, since it is not a peer and does not act in a research capacity. Moreover, even a cursory reading of the ChatGPT suggestions reveals that many of the issues raised by ChatGPT make no sense. For example, the ChatGPT summary suggests that we used "ROCCS" (sic) and RPS probabilistic skill metrics, which are not included in our paper at all. And later, ChatGPT suggests that we also consider using NCEP GEFSv12 because it includes a dynamic sea-level component, which is of course not true (GEFSv13, which is still undergoing development and testing, will be the first operational NCEP FV3-based atmosphere-ocean coupled model). Many of the ChatGPT questions are trivially answered in the paper. For example, it suggests that we only compute ensemble mean skill metrics (which is not true) and then suggests that we do not discuss bias corrections, which is patently false; indeed, our manuscript explicitly discusses both ensemble mean bias corrections (Sect. 2.1) and ensemble spread probability distribution bias correction (Sect. 2.3).